# Circulating myeloid cells invade the central nervous system to mediate cachexia during pancreatic cancer

Kevin G Burfeind[1,2]*, Xinxia Zhu[1], Mason A Norgard[1], Peter R Levasseur[1], Christian Huisman[1], Abigail C Buenafe[1], Brennan Olson[1,2], Katherine A Michaelis[1,2], Eileen RS Torres[3], Sophia Jeng[4,5], Shannon McWeeney[4,5,6], Jacob Raber[3,7], Daniel L Marks[1,5,8]*

[1]Papé Family Pediatric Research Institute, Oregon Health & Science University, Portland, United States; [2]Medical Scientist Training Program, Oregon Health & Science University, Portland, United States; [3]Department of Behavioral Neuroscience, Oregon Health & Science University, Portland, United States; [4]Oregon Clinical and Translational Research Institute, Oregon Health & Science University, Portland, United States; [5]Knight Cancer Institute, Oregon Health & Science University, Portland, United States; [6]Division of Bioinformatics and Computational Biology, Department of Medical Informatics and Clinical Epidemiology, Oregon Health & Science University, Portland, United States; [7]Departments of Neurology and Radiation Medicine, Division of Neuroscience ONPRC, Oregon Health and & Science University, Portland, United States; [8]Brenden-Colson Center for Pancreatic Care, Oregon Health and & Science University Portland, Portland, United States

*For correspondence:
burfeind@ohsu.edu (KGB);
marksd@ohsu.edu (DLM)

Competing interests: The authors declare that no competing interests exist.

**Abstract** Weight loss and anorexia are common symptoms in cancer patients that occur prior to initiation of cancer therapy. Inflammation in the brain is a driver of these symptoms, yet cellular sources of neuroinflammation during malignancy are unknown. In a mouse model of pancreatic ductal adenocarcinoma (PDAC), we observed early and robust myeloid cell infiltration into the brain. Infiltrating immune cells were predominately neutrophils, which accumulated at a unique central nervous system entry portal called the velum interpositum, where they expressed CCR2. Pharmacologic CCR2 blockade and genetic deletion of *Ccr2* both resulted in significantly decreased brain-infiltrating myeloid cells as well as attenuated cachexia during PDAC. Lastly, intracerebroventricular blockade of the purinergic receptor P2RX7 during PDAC abolished immune cell recruitment to the brain and attenuated anorexia. Our data demonstrate a novel function for the CCR2/CCL2 axis in recruiting neutrophils to the brain, which drives anorexia and muscle catabolism.

## Introduction

Cancer patients commonly present with symptoms driven by disruption of normal CNS function. Weight loss, weakness, fatigue, and cognitive decline often occur in malignancies outside the CNS, and develop prior to initiation of cancer therapy (*Meyers, 2000*; *Miller et al., 2008*; *Olson and Marks, 2019*). Many of these symptoms are part of a syndrome called cachexia, a devastating state of malnutrition characterized by decreased appetite, fatigue, adipose tissue loss, and muscle catabolism (*Fearon et al., 2011*). There are currently no effective treatments for cachexia or other CNS-mediated cancer symptoms. While mechanisms of CNS dysfunction during malignancy are still not

**eLife digest** Weight loss, decreased appetite and fatigue are symptoms of a wasting disorder known as cachexia, which is common in several serious diseases such as AIDS, chronic lung disease and heart failure. Up to 80 percent of people with advanced cancer also develop cachexia, and there are no effective treatments.

It is not known how cachexia develops, but symptoms like appetite loss and fatigue are controlled by the brain. One theory is that the brain may be responding to a malfunctioning immune response that causes inflammation. While the brain was thought to be protected from this, new research has shown that it is possible for cells from the immune system to reach the brain in some conditions. To find out if this also happens in cancer, Burfeind et al. studied mice that had been implanted with pancreatic cancer cells and were showing signs of cachexia.

Samples from the mice's brains showed that immune cells known as neutrophils were present and active. A protein known as CCR2 was found in higher levels in the brains of these mice. This protein is involved in the movement of neutrophil cells through the body. To see what effect this protein had, Burfeind et al. gave the mice a drug that blocks CCR2. This prevented the neutrophils from entering the brain and reduced the symptoms of cachexia in the mice.

To further confirm the role of CCR2, the mice were genetically modified so that they could not produce the protein. This reduced the number of neutrophils seen in the brain but not in the rest of the body. This suggests that a drug targeting CCR2 could help to reduce the symptoms of cachexia, without disrupting the normal immune response away from the brain. This approach would still need to be tested in clinical trials before it is possible to know how effective it might be in humans.

well understood, inflammation in the brain is proposed as a key driver (*Burfeind et al., 2016*). Inflammatory molecules (e.g. lipopolysaccharide, cytokines) can cause dysfunction of the appetite-, cognition-, weight-, and activity-regulating regions in the CNS, resulting in signs and symptoms nearly identical to those observed during cancer (*Braun et al., 2011*; *Burfeind et al., 2016*; *Grossberg et al., 2011*). Moreover, cytokines and chemokines are produced in these same regions during multiple types of cancer (*Braun et al., 2011*; *Michaelis et al., 2017*). Our lab and others previously showed that disrupting inflammatory signaling by deleting either MyD88 or TRIF attenuates anorexia, muscle catabolism, fatigue, and neuroinflammation during malignancy (*Burfeind et al., 2018*; *Ruud et al., 2013*; *Zhu et al., 2019*).

The mechanisms by which inflammation generated in the periphery (e.g. at the site of a malignancy) is translated into inflammation in the brain, and how this is subsequently translated CNS dysfunction, are still not known. Circulating immune cells present an intriguing cellular candidate, as they are thought to infiltrate and interact with the brain during various states of inflammation (*Prinz and Priller, 2017*), yet have not been investigated as potential mediators of brain dysfunction during cancer. We utilized a syngeneic, immunocompetent, mouse model of pancreatic ductal adenocarcinoma (PDAC), a deadly malignancy associated with profound anorexia, fatigue, weakness, and cognitive dysfunction (*Baekelandt et al., 2016*; *Michaelis et al., 2017*). We first demonstrated that the inflammatory transcripts upregulated in the CNS during PDAC consisted largely of chemokines. We then characterized the identity, properties, and function of immune cells in the brain during PDAC. We observed that circulating myeloid cells, primarily neutrophils, were recruited to the CNS early in PDAC, infiltrating throughout the brain parenchyma and accumulating in the meninges near regions important for appetite, behavior, and body composition regulation. We then demonstrated that CCR2 signaling is important for immune cell recruitment to the brain and cachexia during PDAC. Next, we blocked purinergic receptor P2RX7 signaling specifically on brain macrophages during PDAC via intracerebroventricular (ICV) injection of oxidized ATP (oATP), which prevented circulating myeloid cell recruitment to the brain and attenuated anorexia. Taken together, these results reveal a novel mechanism by which neutrophil and other myeloid cells are recruitment to the brain, where they contribute to cachexia symptoms.

# Results

## Chemokine transcripts are upregulated in the brain in a mouse model of PDAC

Our lab previously demonstrated that inflammatory cytokine transcripts are upregulated in the hypothalamus in a mouse model of PDAC (*Burfeind et al., 2018*; *Michaelis et al., 2017*; *Zhu et al., 2019*). To our knowledge, no studies have investigated whether inflammatory transcripts are upregulated during cachexia in other brain regions important for behavior and metabolism. Therefore, we used qRT-PCR to determine if various inflammatory cytokine or chemokine transcripts were upregulated in the hippocampus, hypothalamus, or area postrema during PDAC. We utilized a mouse model of PDAC, generated through a single intraperitoneal (IP) or orthotopic (OT) injection of C57BL/6 *Kras*^G12D *Tp53*^R172H *Pdx1*-Cre^+/+ (KPC) cells. This well-characterized model recapitulates several key signs and symptoms of CNS dysfunction observed in humans, including anorexia, muscle catabolism, and fatigue (*Burfeind et al., 2018*; *Michaelis et al., 2019*; *Michaelis et al., 2017*; *Zhu et al., 2019*). We performed qRT-PCR at 10 days post-IP inoculation, a time when animals reliably develop anorexia, muscle catabolism, and decreased locomotor activity (*Michaelis et al., 2017*). We queried transcripts previously demonstrated to be upregulated in the brain during chronic systemic inflammation (*Burfeind et al., 2018*; *Grossberg et al., 2010*). In addition, no studies have investigated chemokine expression in the brain during extra-CNS malignancy, so we also queried expression of chemokine transcripts.

In agreement with our previous studies we observed that *Il-1β* was upregulated in the hypothalamus (*Figure 1*). It was also upregulated in the area postrema, and showed a trend toward significance in the hippocampus (p=0.08). However, of the other cytokine transcripts analyzed, only those coding for prostaglandin synthase D2 (*Ptgs2* – in the hypothalamus and area postrema, but not the hippocampus) and IL-1R (*Il1r* - again in the hypothalamus and area postrema, but not the

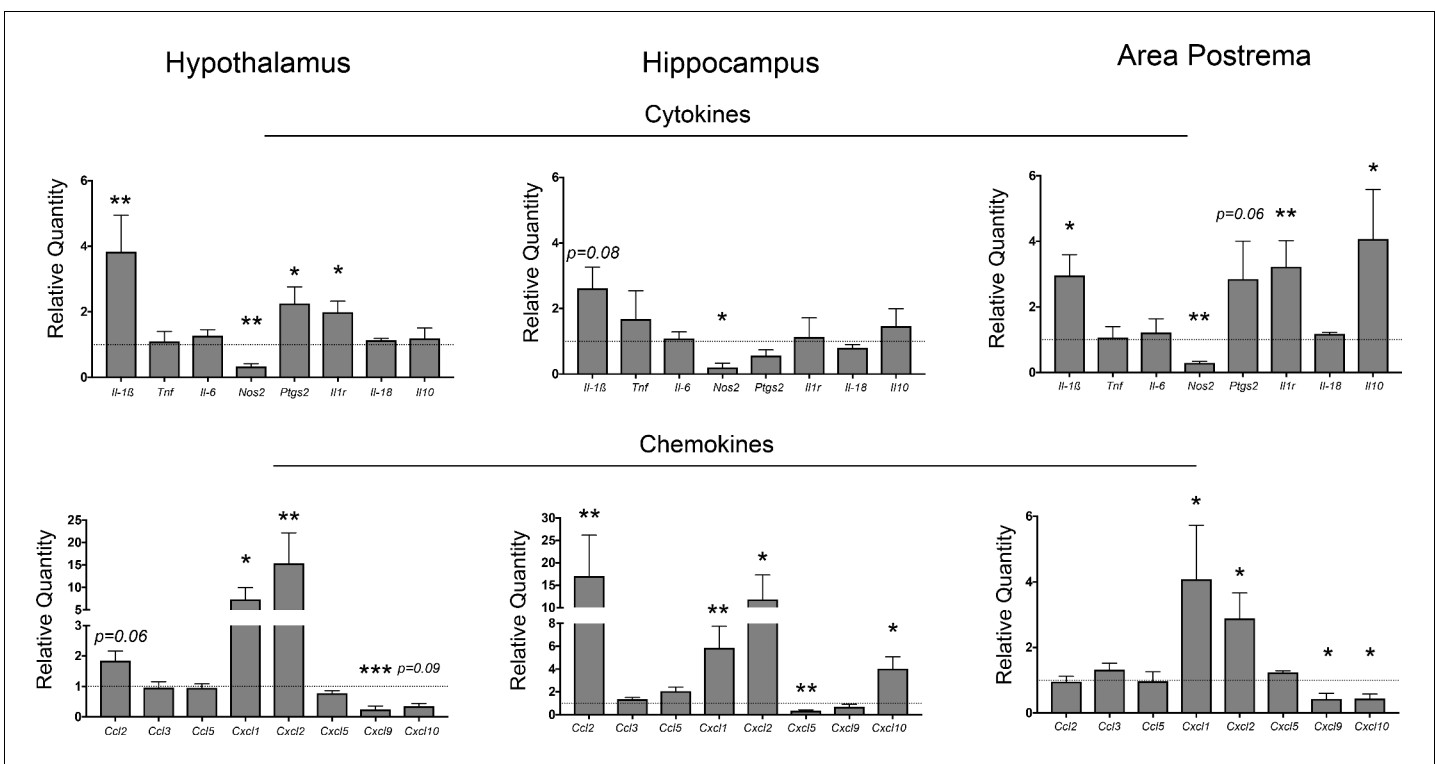

**Figure 1.** Neuroinflammation in the CNS during PDAC. qRT-PCR analysis of cytokine and chemokine transcripts in the hypothalamus, hippocampus, and area postrema in PDAC-bearing animals at 10 d.p.i. Values are relative to sham group. All analyses are from 10 d.p.i. *n* = 4–5/group, *p<0.05, **p<0.01, ***p<0.001 compared to sham group in one-way ANOVA analysis of ΔCT values. Results are representative of at least two independent experiments. For all figures, data are presented as mean ± s.e.m.

hippocampus) were upregulated. The anti-inflammatory transcript *Il10* was upregulated in the area postrema only. Interestingly, the transcript coding for nitric oxide synthase 2 (*Nos2* – induced during inflammation and mainly expressed by endothelial cells) was downregulated in all three brain regions.

Several chemokine transcripts associated with myeloid cell chemotaxis were upregulated in different brain regions during PDAC. Two of the three *IL-8* orthologues, *Cxcl1* and *Cxcl2*, were highly upregulated in all three regions investigated. *Ccl2* was highly upregulated in the hippocampus, and nearly significantly upregulated in the hypothalamus (p=0.06). Alternatively, *Cxcl9* was downregulated in both the area postrema and hypothalamus, whereas *Cxcl10* was downregulated in the area postrema, yet upregulated in the hippocampus. Lastly, the third IL-8 orthologue, *Cxcl5*, was downregulated in the hippocampus.

## Circulating myeloid cells infiltrate the brain early in PDAC

Based on our observation that there is robust upregulation of several chemokine transcripts in the CNS in our mouse model of cachexia, along with our previous data showing that the transcript for the leukocyte adhesion molecule, P-selectin is upregulated in the brain during cachexia (*Michaelis et al., 2017*), we hypothesized that immune cells infiltrated the brain. We utilized flow cytometry to perform an initial brain-wide analysis of infiltrating immune cells in our PDAC model. Using 10-color flow cytometry of whole brain homogenate (*Figure 2A*), we characterized brain immune cells at three time points: 5 days post-inoculation (d.p.i) (before anorexia, fatigue, and muscle mass loss onset), 7 d.p.i. (initiation of wasting and anorexia), and 10 d.p.i. (robust wasting and anorexia, but 4–5 days before death) after IP injection of PDAC cells (see Figure 6F for typical disease progression of our KPC model). Compared to sham-injected animals, we observed a significant increase in CD45$^{high}$CD11b+ myeloid cells in the brains of animals with PDAC (*Figure 2B*), with an increase as a percentage of total CD45+ (all immune cells) and CD45$^{high}$ (non-microglia leukocytes) cells occurring at 5 d.p.i. (*Figure 2D* and *Figure 2—figure supplement 1D*).

Both absolute and relative number of lymphocytes (CD45$^{high}$CD11b-) were decreased in the brains of tumor animals compared to sham animals starting at 5 d.p.i., which was driven by a decrease in B-cells and CD4+ T-cells (*Figure 2C* and *Figure 2—figure supplement 1B–D*). There was no change in number of microglia (defined as CD45$^{mid}$CD11b+) throughout the disease course (*Figure 2C*). Further phenotypic analysis of infiltrating myeloid cells revealed that by 7 d.p.i., there was an increase in relative number (as a percentage of total CD45+ and CD45$^{high}$) of Ly6C$^{mid}$Ly6-G$^{high}$ neutrophils, Ly6C$^{low}$ myeloid cells, and Ly6C$^{high}$ monocytes (*Figure 1D,E* and *Figure 2—figure supplement 1D*). We observed an increase in absolute number of neutrophils, Ly6C$^{high}$ monocytes, and Ly6C$^{low}$ myeloid cells starting at 7 d.p.i., which became significant at either 7 (Ly6C$^{high}$ monocytes) or 10 d.p.i. (neutrophils and Ly6C$^{low}$ myeloid cells) (*Figure 2D and E*). Neutrophils were by far the most numerous invading myeloid cell type, constituting 34% percent of CD45$^{high}$CD11b+ cells in sham animals, and increasing to nearly 54% by 10.d.p.i. in tumor animals (*Figure 2F*).

In order to verify that the population, we defined as 'microglia' were actually microglia and also confirm that Ly6G+CD45$^{high}$ myeloid cells were neutrophils and not an artifact of nonspecific antibody binding on microglial or other brain macrophages, we performed an additional flow cytometry experiment incorporating CX3CR1 (a marker of macrophages). We observed that 95% of the population we defined as 'microglia' expressed CX3CR1, while only 6% of the population we defined as 'neutrophils' expressed this protein, confirming the identity of these cells as microglia and neutrophils, respectively (*Figure 2—figure supplement 2*).

We considered the population within the gate labeled 'CD45$^{high}$ myeloid cells' to be mainly infiltrating immune cells. While we drew this gate based on a clearly defined population of the cells (see right panel on *Figure 2B*), which we demonstrated consisted mainly of neutrophils (*Figure 2F*), it is possible that the population within this gate could also contain activated microglia. Furthermore, the population of CD45$^{high}$CD11b+Ly6C$^{low}$ myeloid cells could also be activated microglia. To address these issues, we generated GFP+ bone marrow chimera mice through conditioning WT mice with treosulfan to ablate marrow, then transplanting marrow from pan-GFP mice (Ly5.1$^{GFP}$) (*Figure 2—figure supplement 3A*). This system is advantageous because, unlike other alkylating agents, treosulfan does not cross or disrupt the blood brain barrier (*Capotondo et al., 2012*). On average, mice that underwent bone marrow transplant (GFP BMT mice) exhibited 75% chimerism (*Figure 2—figure supplement 3C*). In agreement with results from WT marrow animals, we observed that at 10 d.p.i.,

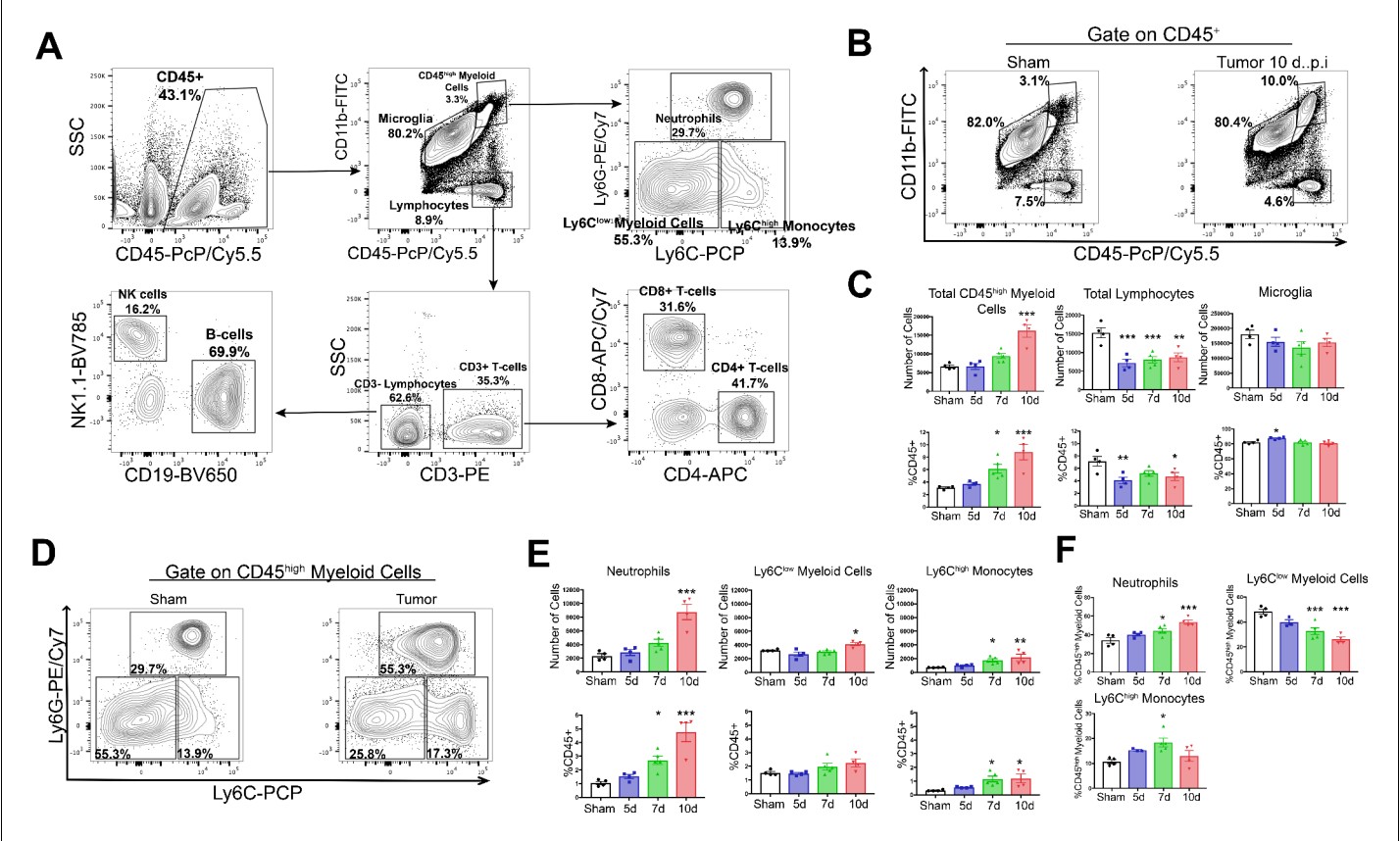

**Figure 2.** Circulating myeloid cells infiltrate the brain during PDAC. (**A**) Flow cytometry plots of immune cells isolated from whole brain homogenate, showing gating strategy to identify different immune cell populations. (**B**) Representative flow cytometry plots displaying CD45 and CD11b fluorescent intensities of immune cells isolated from brains of tumor and sham animals, gated on live, singlet, CD45+ cells. (**C**) Quantification of different immune cell populations in the brain at different time points throughout PDAC course. d = days post inoculation. Populations were identified as shown in (**A**). (**D**) Representative flow cytometry plots displaying Ly6C and Ly6G fluorescent intensities of immune cells isolated from brains of tumor and sham animals, gated on CD45highCD11bhigh cells. (**E**) Quantification of different CD45high myeloid cell populations in the brain at different time points during PDAC progression. (**F**) Relative amounts of different CD45high myeloid cell populations as a percentage of total CD45high myeloid cells, throughout the course of PDAC. Populations identified as described for (**E**). n = 4–5/group, *p<0.05, **p<0.01, ***p<0.001 compared to sham group in one-way ANOVA Bonferroni *post hoc* analysis, and results are representative of three independent experiments.

The online version of this article includes the following figure supplement(s) for figure 2:

**Figure supplement 1.** Decreased lymphocytes in the brain during PDAC cachexia.

**Figure supplement 2.** Infiltrating Ly6G+ cells are not microglia.

**Figure supplement 3.** GFP BMT confirms peripheral origin of infiltrating myeloid cells in the CNS during PDAC.

thousands of GFP+ myeloid cells infiltrated the brain in tumor animals (*Figure 2—figure supplement 3B and D*). The majority of these cells were neutrophils, with a concurrent increase in Ly6Chigh monocytes (*Figure 2—figure supplement 3C-F*). As we observed previously, this coincided with a decrease in brain lymphocytes (CD45+GFP+CD11b-) in tumor animals (Figure 2—figure supplement 3D). We did not observe an increase in GFP+ Ly6Clow myeloid cells (Figure 2—figure supplement 3F), suggesting that the increase in CD45highCD11b+Ly6Clow cells in our WT marrow PDAC mice was a result of microglia activation, rather than infiltrating monocytes.

Taken together, these data show that myeloid cells infiltrate the brain during PDAC, temporally correlating with symptom onset. The majority of infiltrating immune cells were neutrophils. Since the purpose of this study was to investigate infiltrating cells, we chose to focus our subsequent analysis on myeloid cells, with an emphasis on neutrophils.

## Invading myeloid cells accumulate at CNS interfaces during PDAC

Prior studies demonstrated regional vulnerability in the CNS to immune cell invasion during systemic inflammation (*D'Mello et al., 2009*). Therefore, we investigated the anatomic distribution of infiltrating myeloid cells in the CNS during PDAC. We performed immunofluorescence immunohistochemistry analysis at 10 d.p.i. in IP-inoculated animals, since all tumor-inoculated animals reliably developed anorexia, fatigue, and muscle catabolism at this time point, yet were not at terminal stage (*Michaelis et al., 2017*). In addition, our flow cytometry analysis demonstrated a robust immune cell infiltrate in the brain at 10 d.p.i. For initial analysis, we defined leukocytes as CD45+ globoid cells. Although we observed scattered CD45+ globoid cells within the parenchyma in the cortex and thalamus in tumor mice (*Figure 3—figure supplement 1*), we observed a robust increase in leukocytes in the meninges adjacent to the hippocampus and median eminence (ME) (*Figure 3B and C*). We also performed quantification in the area postrema, based on our previous experiment demonstrating chemokine transcript upregulation in this region. While there was an increase in overall CD45 immunoreactivity in the area postrema, these cells appeared ramified rather than globoid (*Figure 3D*), suggesting microglia activation rather than immune cell infiltration. We did not observe any CD45+ cells in the lateral parabrachial nucleus (data not shown), which was implicated in cancer-associated anorexia (*Campos et al., 2017*). This was perhaps due to its lack of proximity to a circumventricular organ or meninges. Interestingly, we observed an increase in neutrophils (defined as myeloperoxidase [MPO] positive, CD45+ globoid cells) only in the meninges surrounding the hippocampus (*Figure 3B*). This layer of meninges, known as the velum interpositum (VI), is a double-layered invagination of the pia matter. This potential space is closed rostrally, communicates

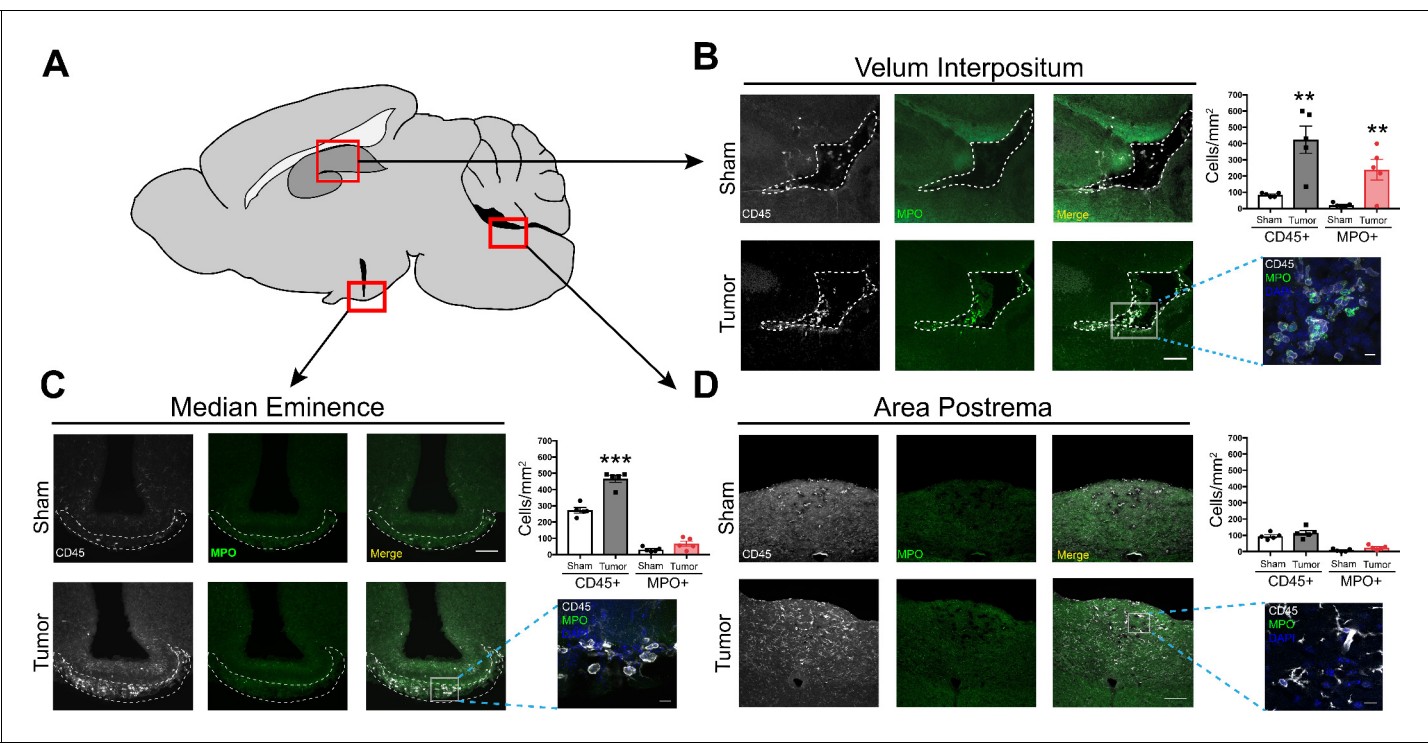

**Figure 3.** Infiltrating immune cells accumulate at CNS interfaces during PDAC cachexia. (**A**) Picture of sagittal mouse brain section to illustrate different regions analyzed. (**B-D**) 20X images of velum interpositum (**B**), mediobasal hypothalamus (**C**), and area postrema (**D**) of brain from sham animal and tumor animal at 10 d.p.i., with 60X inset shown on the right, along with quantification of MPO+ and total CD45+ cells. For B, dashed line denotes VI borders. For C, dashed line denotes borders of meninges adjacent to ME. Scale bar for 20X images = 100 μm. Scale bar for 60X insets = 10 μm. Data are presented as mean ± s.e.m., *n* = 5/group, *p<0.05, **p<0.01 in student's t-test.

The online version of this article includes the following figure supplement(s) for figure 3:

**Figure supplement 1.** Immunofluorescence analysis of infiltrating immune cells during PDAC.

**Figure supplement 2.** Characteristics of brain-infiltrating immune cells during PDAC.

**Figure supplement 3.** Neutrophils are not depleted by Ly6G antibody administration.

caudally with the quadrigeminal cistern, and is highly vascularized via a number of internal cerebral arterioles and veins. Recent studies demonstrate robust immune cell recruitment into the brain via this anatomical route after mild trauma, during CNS infection, and during CNS autoimmune disease (*Alvarez and Teale, 2006*; *Schmitt et al., 2012*; *Szmydynger-Chodobska et al., 2016*).

We verified the presence of meninges in the VI with ER-TR7 labeling, which showed infiltrating neutrophils in the VI meninges in tumor mice (*Figure 3—figure supplement 2A*). Neutrophils in the VI were degranulating, with MPO 'blebs' present at the edge of many cells, along with extracellular MPO (*Figure 3—figure supplement 2C*). This phenomenon was only present in brains of tumor animals and not in brains of sham animals. We were able to confirm neutrophil identity with the plasma membrane marker Ly6G and globoid morphology (*Figure 3—figure supplement 1C*). Neutrophil extracellular traps (NETs) were also present in the VI, as identified by citrillunated histone H3 and MPO co-labeling (*Figure 3—figure supplement 2E*). We were unable to perform quantification on the number of NETs present in tumor mouse brains, due to the transient nature of these events.

In the CNS parenchyma, especially in the thalamus and cortex, we frequently observed neutrophils undergoing phagocytosis by microglia, with Iba-1+ cells extending processes around MPO+ neutrophils (*Figure 3—figure supplement 2B*). This supports previous studies showing that microglia protect the CNS parenchyma from neutrophil invasion during various states of inflammation (*Neumann et al., 2018*; *Neumann et al., 2008*; *Otxoa-de-Amezaga et al., 2019*).

The peripheral origin of the CD45+ globoid cells in the brain was assessed using our GFP BMT mice. Sham BMT mice showed very few GFP+ cells in the brain, including the cortex and thalamus (*Figure 3—figure supplement 1A*), as well as the meninges (data not shown). In contrast, there was a large increase in GFP+ cells in the brains of KPC mice at 10 d.p.i. We observed a pattern of infiltrating GFP+ cells that was identical to CD45+ globoid cells in our previous experiments, with scattered GFP+ cells in the cortex and thalamus (*Figure 3—figure supplement 1A and B*), and accumulations of GFP+ cells in the VI (*Figure 3—figure supplement 2D*). In agreement with our previous data, GFP+ cells were MPO+ in the VI (*Figure 3—figure supplement 2D*).

Since neutrophils were the predominant cell type infiltrating the brain during PDAC, we hypothesized that these cells are key drivers of cachexia. To test this hypothesis, we attempted to deplete neutrophils in our mouse model of PDAC. Beginning at 2 d.p.i., we treated tumor-bearing animals with either 500 µg anti-Ly6G antibody (clone 1A8) or isotype control IgG daily (*Figure 3—figure supplement 3A*). At 10 d.p.i., there was complete abrogation of the population we defined as neutrophils (CD45+CD11b+Ly6C$^{mid/high}$Ly6G$^{high}$) in the circulation of 1A8-treated animals (*Figure 3—figure supplement 3D and E*). However, there was a suspicious population of CD45+CD11b+Ly6C$^{mid}$Ly6G- cells in 1A8-treated animals, which was not present in the isotype-treated animals, and had identical forward scatter and side scatter properties as neutrophils (*Figure 3—figure supplement 3C and D*). Since we used a fluorescently labeled anti-Ly6G antibody to identify neutrophils, it is possible that the neutralizing Ly6G antibody bound to all of the Ly6G antigens on neutrophils, therefore preventing the fluorescently labeled Ly6G antibody from binding. To address this issue, we used additional markers to identify neutrophils which did not require Ly6G labeling. It was previously reported that neutrophils could be differentiated from other circulating myeloid cells with the markers Dec205 and CD115 (*Napier et al., 2015*). We identified a population of cells that was CD45+CD11b+Dec205+CD115-, which was 95% Ly6C$^{mid/high}$Ly6G+ neutrophils (*Figure 3—figure supplement 3F*). Using this new definition for neutrophils, we observed that there was no decrease in this population at 10 d.p.i. after daily treatment with 500 µg anti-Ly6G antibody (*Figure 3—figure supplement 3E*).

These results suggest that during PDAC, chronic neutrophil depletion with anti-Ly6G antibody is not possible.

## CCR2 inhibition attenuates anorexia and immune cell infiltration into the velum interpositum during PDAC

To identify unique mechanisms of immune cell recruitment to the brain and determine if inhibiting immune cells from infiltrating the brain attenuates cachexia, we treated OT-implanted tumor-bearing mice with either a CCR2 inhibitor (RS504393, Tocris) or CXCR2 inhibitor (SB225002, Tocris). We chose CCR2 and CXCR2 since *Ccl2* (coding for the ligand for CCR2), *Cxcl1* (which codes for CXCL1, a ligand for CXCR2), and *Cxcl2* (which codes for CXCL2, also a ligand for CXCR2) were the most upregulated chemokine genes in dissected hippocampi (which also included the VI) during PDAC

(*Figure 1*). Furthermore, these are the key chemokines for monocyte and neutrophil chemotaxis, which were the predominant cell types that infiltrated the brain in our PDAC mouse model (*Figure 2*).

RS504393 and SB225002 were previously demonstrated to be highly effective and specific small-molecule inhibitors of their respective receptors (*Nywening et al., 2018*). Based on dosing regimens optimized previously (*Nywening et al., 2018*), we administered 5 mg/kg RS504393, 10 mg/kg SB225002, or vehicle (DMSO) subcutaneously twice daily starting at 3 d.p.i. (*Figure 4A*). We used immunofluorescence analysis to quantify total CD45+ globoid cells and MPO+ cells in the VI in vehicle-, RS504393-, and SB225002-treated tumor-bearing animals. We focused our initial analysis on the VI, as it was a key region for invading immune cell accumulation. We observed a decrease in CD45+ globoid cells in the VI in RS504393-treated tumor-bearing animals compared to vehicle-treated tumor-bearing animals (*Figure 4B and C*). Alternatively, while there was a slight decrease in CD45+ cells in the VI in SB225002-treated tumor-bearing animals compared to vehicle-treated tumor-bearing animals, this difference was not significant (*Figure 4D*). Compared with vehicle-treated tumor-bearing animals, there was a moderate decrease in MPO+ cells in the VI in both SB225002- and RS504393-treated tumor-bearing animals, but this difference was also not significant.

While all groups consumed similar amounts of food over the first 7 days of the study (pre-cachexia) (*Figure 4—figure supplement 1A*), RS504393-treated animals experienced decreased anorexia, as evidenced by increased food intake, compared to vehicle-treated tumor-bearing animals (*Figure 4E*), Alternatively, SB225002-treated tumor-bearing had similar food intake compared to vehicle-treated animals. RS504393-treated animals had a nonsignificant increase in gastrocnemius mass compared to vehicle-treated animals, but did have an increase in heart mass compared to vehicle-treated animals, indicating decreased muscle catabolism (*Figure 4F and G*). There was no difference in heart or gastrocnemius mass in SB225002-treated animals compared to vehicle-treated animals. There was also no difference in tumor mass, or circulating immune cells in SB225002- or RS504393-treated animals compared to vehicle-treated animals (*Figure 4—figure supplement 1B and C*).

These data demonstrate that pharmacologic inhibition of CCR2 is important for cachexia and immune cell recruitment to the brain during PDAC. Therefore, we next performed additional studies to further characterize the activity of the CCL2/CCR2 axis in the brain.

## The CCR2-CCL2 axis is activated in the CNS during PDAC

Using in situ hybridization we localized robust CCL2 mRNA expression exclusively within the VI during PDAC. There was no observable *Ccl2* mRNA in the brains of sham animals (*Figure 5A*). We verified these results at the protein level using *Ccl2*[mCherry] mice, which showed abundant CCL2 protein expression in the VI in tumor animals at 10 d.p.i., exclusively expressed in Iba1+CD206+ meningeal macrophages. CCL2 protein was not expressed in VI meningeal macrophages in sham mice (*Figure 5B*). We did not observe robust CCL2 protein expression in any other locations in the brain.

*Ccr2*[RFP/WT] reporter mice were used to localize CCR2+ cells in the CNS. We observed that, at 10 d.p.i., CCR2+ immune cells infiltrated the brains of tumor mice and accumulated in the VI (*Figure 5C*) Interestingly, a large percentage of neutrophils in the VI were CCR2+ (*Figure 5D*), which infiltrated throughout the VI and often formed large aggregates consisting of 20 cells or more (*Figure 5—figure supplement 1A*). CCR2+ cells were sparsely scattered within the surrounding brain parenchyma in tumor mice, whereas none were ever observed in sham controls.

In order to verify CCR2 expression on neutrophils in the brains of tumor-bearing animals, we performed flow cytometry for CCR2 (using an anti-CCR2 antibody) on Ly6G+ circulating, liver-infiltrating, and brain-infiltrating neutrophils in both sham and PDAC-bearing animals at 10 d.p.i. As expected, we observed minimal CCR2 expression on circulating neutrophils in sham animals. While there was a slight increase in circulating CCR2+ neutrophils in tumor-bearing animals, there was no increase in CCR2+ neutrophils in the liver. Alternatively, there was a large increase in CCR2+ neutrophils in the brains of tumor-bearing animals (*Figure 5—figure supplement 1B*).

## Tumor-derived factors induce CCL2 production in brain macrophages

Since we observed robust production of CCL2 in meningeal macrophages during PDAC, we hypothesized that tumor-derived factors could induce CCL2 or other chemokines in brain macrophages. To

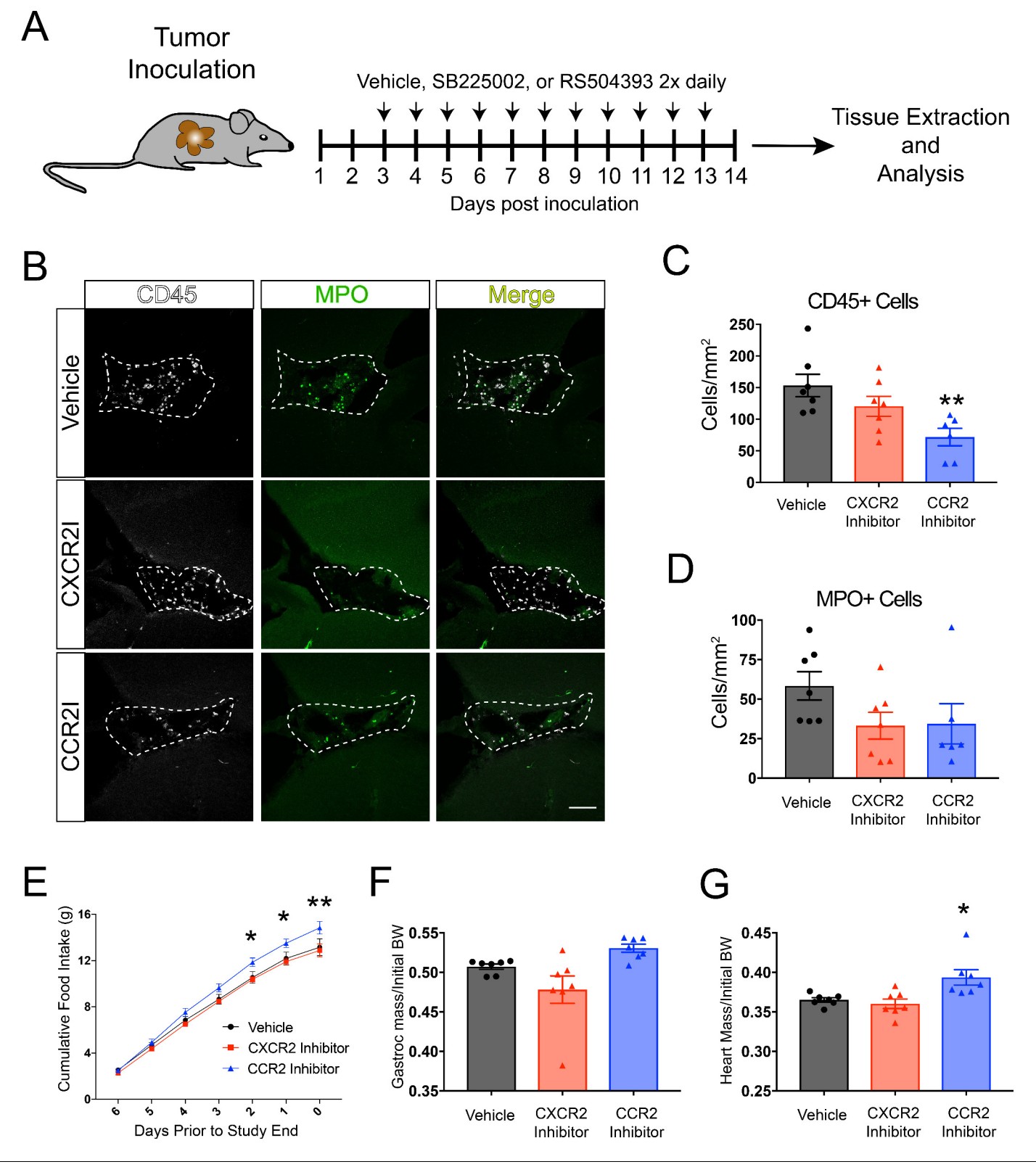

**Figure 4.** CCR2 signaling is important for cachexia and immune cell infiltration into the brain during PDAC. (**A**) Diagram depicting treatment schedule after OT tumor inoculation with PDAC cells. (**B**) Representative images of the VI from brains of vehicle-, SB225002-, or RS504393-treated tumor-bearing animals at 14 d.p.i. CXCR2I = SB225002. CCR2I = RS504393. Dashed line denotes VI borders. Scale bar = 100 μm. (**C**) Quantification of CD45+ globoid cells in the VI at 14 d.p.i. n = 7/group. **p<0.01 compared to vehicle-treated in Bonferroni post-hoc analysis in one-way ANOVA. (**D**) Quantification of

*Figure 4 continued on next page*

*Figure 4 continued*

MPO+ cells in the VI at 14 d.p.i. One RS504393-treated animal was excluded from quantification analysis due to meeting Grubbs outlier criterion. (**E**) Cumulative food intake starting when animals develop cachexia, at 7 d.p.i. *p<0.05, **p<0.01, comparing RS504393-treated tumor vs. vehicle-treated tumor in Bonferroni *post hoc* analysis in two-way ANOVA. *n* = 7/group. (**F**) Mass of dissected gastrocnemius, normalized to initial body weight, at 14 d. p.i. (**G**) Mass of dissected heart, normalized to initial body weight, at 14 d.p.i. *p<0.05 compared to vehicle-treated in Bonferroni post-hoc analysis in one-way ANOVA.

The online version of this article includes the following figure supplement(s) for figure 4:

**Figure supplement 1.** No differences in pre-cachexia food intake, tumor mass, or circulating immune cells in SB225002-, or RS504393-treated tumor-bearing animals.

test this hypothesis, we utilized an in vitro microglia culture system used previously by our laboratory (*Burfeind et al., 2020*; *Zhu et al., 2016*). Briefly, mixed glia were isolated from 3-day-old mouse pups, then after 14–16 days of culture, microglia were removed through shaking, were replated, and treated with PDAC conditioned media or control media (FBS+RPMI) (*Figure 6A*). Sixteen hours later, RNA was isolated and qRT-PCR was performed. Chemokine transcripts differentially regulated in more than one brain region in our PDAC mouse model (as shown in *Figure 1*) were selected for analysis. We observed that, of the chemokine transcripts queried, *Ccl2* was by far the most robustly upregulated (approximately 135-fold). *Cxcl1* and *Cxcl2* were upregulated, but not nearly as robustly (15 and approximately 35-fold, respectively) (*Figure 6B*). Alternatively, *Cxcl9* and *Cxcl10* were not induced. 10 ng LPS was used as a positive control, which induced robust upregulation all five chemokine transcripts (*Figure 6C*). These results demonstrate that PDAC-derived factors induce chemokine expression in brain macrophages, and, unlike LPS, there is preferential upregulation of certain chemokines, especially CCL2.

## CCR2 is critical for myeloid cell accumulation at CNS interfaces, anorexia, and muscle catabolism during PDAC

Since small molecule inhibitors can have off-target effects, we used CCR2 knockout (CCR2KO) mice to verify and expand upon our findings demonstrating that CCR2 inhibition is important for cachexia and immune cell infiltration into the brain during PDAC. Using flow cytometry, we observed that at 11 d.p.i. after IP inoculation, there was a 37% decrease in total CD45$^{high}$ myeloid cells in the brains of CCR2KO tumor mice compared to WT tumor mice (*Figure 7A and B*). This difference was driven by a large decrease in brain-infiltrating neutrophils and Ly6C$^{high}$ monocytes. There was also decrease in neutrophils and Ly6C$^{high}$ monocytes as a percentage of CD45$^{high}$ cells in the brains of CCR2KO tumor mice, indicating that differences were not due to a global decrease in infiltrating immune cells (*Figure 7B*). This was also supported by the fact that there were no differences in microglia (data not shown), Ly6C$^{low}$ monocytes, or T-cells in the brains of CCR2KO tumor mice compared to WT tumor mice (*Figure 7C*).

Since CCR2+ immune cells, particularly neutrophils, localized primarily to the VI, we hypothesized that there would be a decrease in immune cells in the VI in CCR2KO tumor animals. Indeed, we observed a dramatic decrease in both total CD45+ globoid and MPO+ immune cells in the VI in CCR2KO tumor mice compared to WT tumor mice (*Figure 7D and E*).

In agreement with our findings from pharmacologic inhibition of CCR2, we observed that CCR2KO mice had decreased anorexia during PDAC compared to WT tumor mice (*Figure 7F*). CCR2KO tumor mice also had attenuated muscle loss compared to WT tumor mice (*Figure 7G*). To determine whether the decreased muscle mass loss in CCR2KO mice was due to decreased muscle proteolysis, we assessed levels of transcripts key for muscle proteolysis in the gastrocnemius, including *Fbxo32* (*Mafbx*), *Trim63* (*Murf1*), and *Foxo1*, which we previously showed are induced by CNS inflammation (*Braun et al., 2011*). We observed that, compared to WT tumor animals, CCR2KO tumor animals had decreased induction of *Murf1* and *Foxo1* (*Figure 7H*), confirming that there was decreased catabolic drive in CCR2KO tumor mice.

Since CCR2 deletion was not brain specific in the CCR2KO mice, we performed an extensive analysis of infiltrating immune cells in other organs (*Figure 7—figure supplement 1*). We observed minimal changes in immune cell composition in the blood, liver, and tumor in CCR2KO tumor mice compared to WT tumor mice. We only observed a decrease in circulating Ly6C$^{high}$ monocytes in

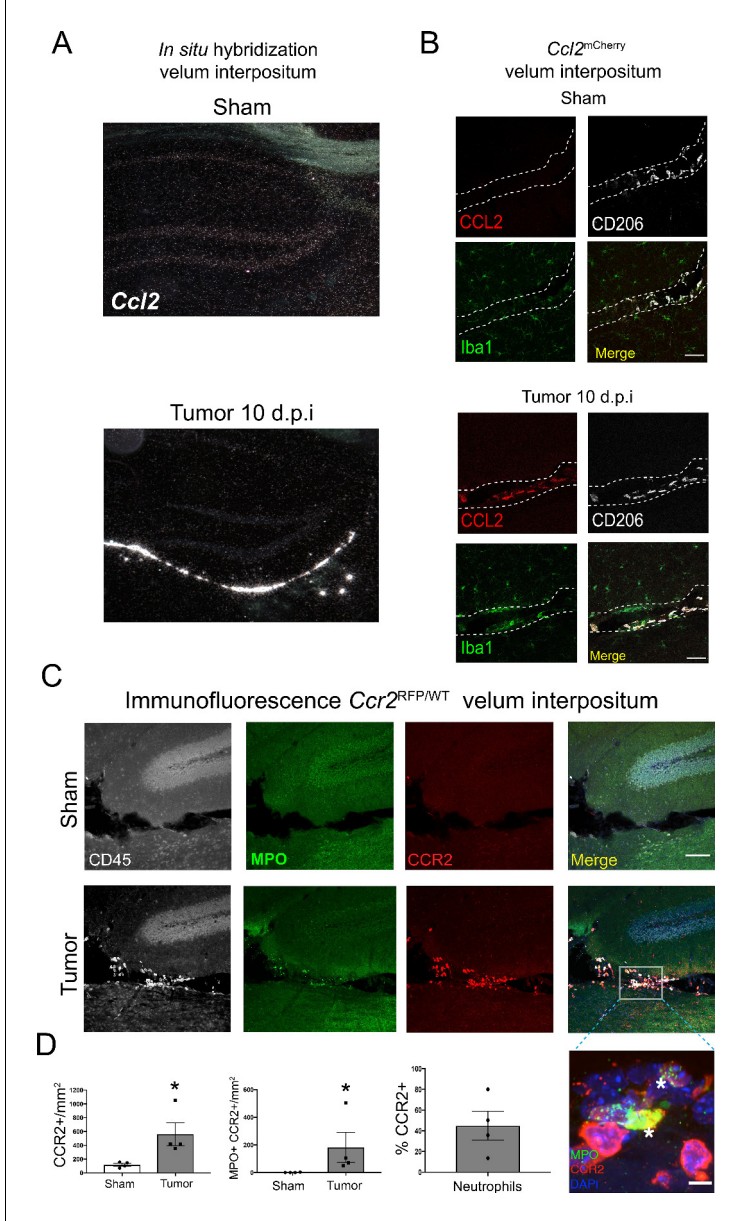

**Figure 5.** The CCR2-CCL2 axis is activated in the CNS during PDAC. (**A**) Representative darkfield microscopy image of in situ hybridization for *Ccl2* in sham and tumor (10 d.p.i) mouse brains. (**B**) Representative 40X confocal microscopy images of the VI from *Ccl2*^mCherry sham and tumor mouse brains, 10 d.p.i., demonstrating that CCL2 protein expression is confined to meningeal macrophages, identified by CD206 labeling. Scale bar = 20 µm. (**C**) Representative 20X confocal microscopy image of brain from *Ccr2*^RFP/WT tumor (10 d.p.i.) and sham mouse brain. Scale bar = 100 µm. Inset = 60X image identifying CCR2+ neutrophils in the VI of a tumor animal, indicated by asterisks. Scale bar = 5 µm. (**D**) Quantification of different RFP+ cell populations in the VI of *Ccr2*^RFP/WT tumor (10 d.p.i.) and sham animals. *n* = 4/group. *p<0.05, Mann-Whitney U-test comparing sham to tumor. Results are representative of two independent experiments.

The online version of this article includes the following figure supplement(s) for figure 5:

**Figure supplement 1.** Neutrophils in the velum interpositum express CCR2 during PDAC.

CCR2KO tumor mice (*Figure 7—figure supplement 1G*). When we assessed neutrophils in CCR2KO tumor mice in different organs as a percentage of those in WT tumor mice, we found the largest decrease to be in the brain and observed a slight increase in circulating neutrophils in CCR2KO tumor mice compared to WT tumor mice (*Figure 7—figure supplement 1F and H*), suggesting that

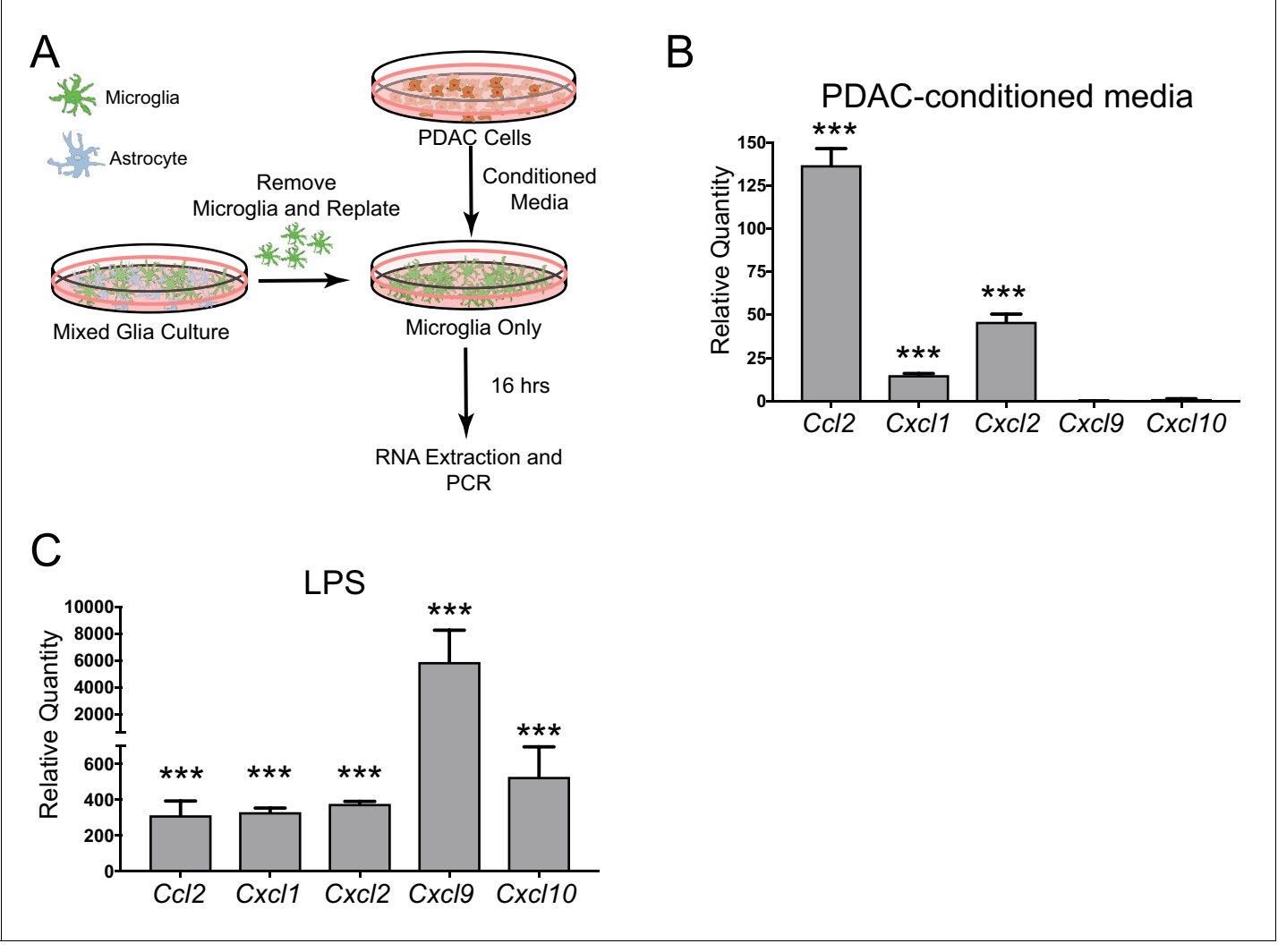

**Figure 6.** Chemokine transcripts are upregulated in microglia in vitro in response to PDAC-conditioned media. (A) Schematic representation of in vitro PDAC-conditioned media treatment system. (B) qRT-PCR analysis of chemokine transcripts after PDAC-conditioned media treatment. Values are relative to those from control media-treated primary microglia. ***p<0.001 compared to control media-treated in repeated measures one-way ANOVA. n = 3/group. Note: one Cxcl9 PDAC-conditioned sample was not amplified after 45 cycles, so was not included. Therefore, no statistics were performed for Cxcl9. (C) qRT-PCR analysis of chemokine transcripts after 10 ng LPS treatment. Values are relative to those from control media-treated primary microglia. ***p<0.001 compared to control media-treated in repeated measures one-way ANOVA. n = 3/group.

the decrease in brain-infiltrating neutrophils was due to a homing defect rather than inability to mobilize from the bone marrow. Therefore, our data show that CCR2 is important for neutrophil recruitment specifically to the brain, and that the decrease in brain-infiltrating neutrophils was due to a homing defect, rather than inability to mobilize from the bone marrow.

Based on our data showing that CCR2 is a brain-specific chemotactic receptor for neutrophils during PDAC, we hypothesized that brain-infiltrating neutrophils are unique compared to neutrophils that infiltrate other organs. In order to characterize the phenotype of brain-infiltrating neutrophils during PDAC, we performed RNA sequencing (RNAseq) on FACS-sorted neutrophils from the blood, liver, tumor, and brain during PDAC at 10 d.p.i, as well as circulating neutrophils from sham animals (*Figure 7—figure supplement 2A and F*). Principal component analysis of individual samples based on the top 500 most varying transcripts revealed that brain-infiltrating neutrophils clustered tightly together, but were distinct from those in the liver, tumor, and blood (*Figure 7—figure supplement 2B*). Furthermore, we were able to identify over 100 transcripts that were

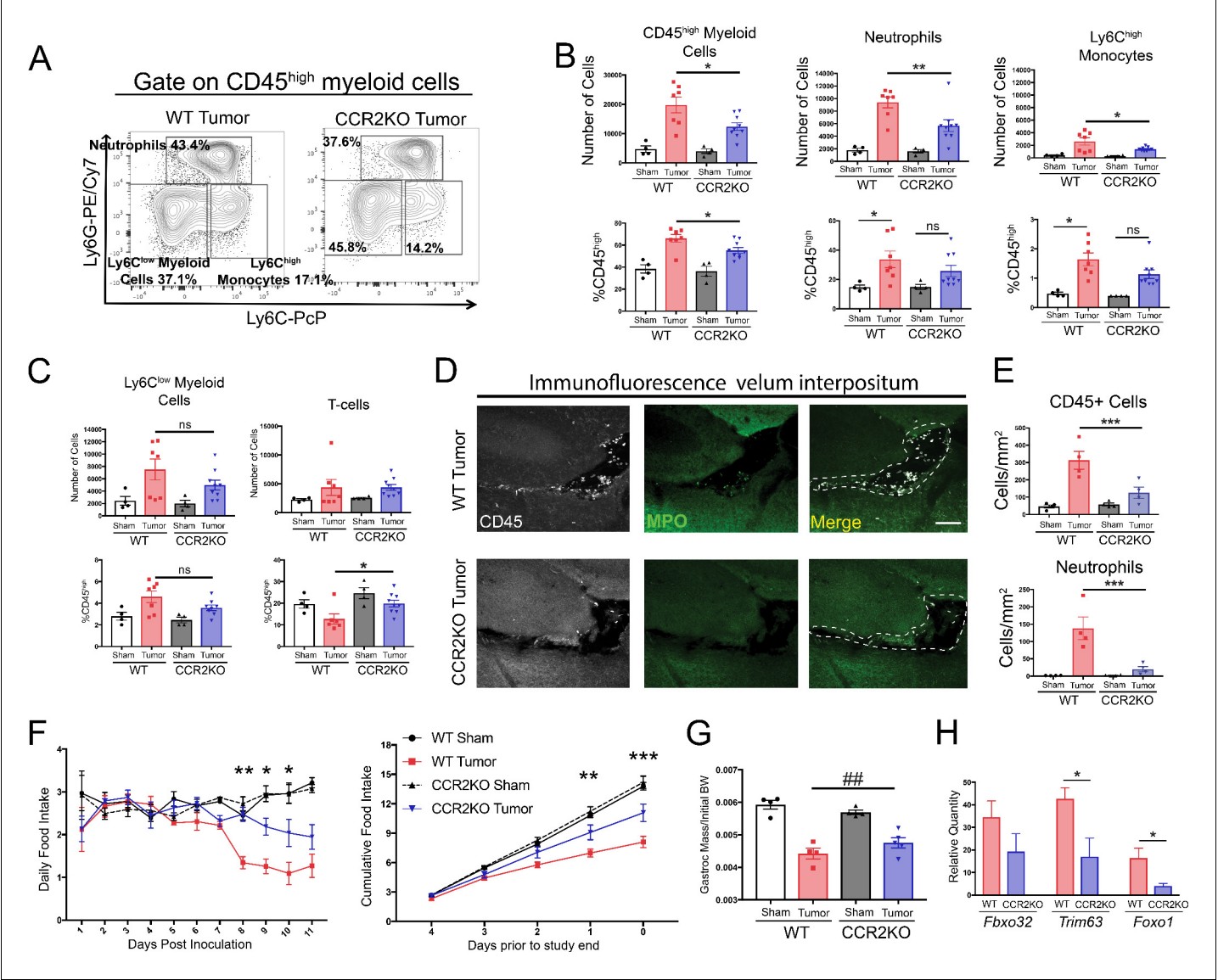

**Figure 7.** The CCR2-CCL2 axis in the CNS is critical for brain inflammation, anorexia, and muscle catabolism during PDAC. (**A**) Representative plot of different CD45[high] myeloid cell populations from WT and CCR2KO tumor animal brains, 11 d.p.i. Cells are gated on live, singlet, CD45+, CD45[high]CD11b+ cells. (**B and C**) Flow cytometry analysis of immune cells isolated from whole brain homogenate. *p<0.05, **p<0.01, WT tumor vs. CCR2KO tumor, or tumor vs. sham in the same genotype in Bonferroni *post hoc* analysis in two-way ANOVA. ns = not significant. *n* = 4–9/group. Data consist of two independent experiments pooled (*n* = at least 2/group in each experiment). (**D**) Representative 20X confocal microscopy images of the VI from WT tumor and CCR2KO tumor brain, 10 d.p.i. Scale bar = 100 μm. Dashed lines represent VI boundary. (**E**) Quantification of total CD45+ globoid cells and MPO+ neutrophils in the VI of WT and CCR2KO animals, 10 d.p.i. ***p<0.001, WT tumor vs. CCR2KO tumor in Bonferroni *post hoc* analysis in two-way ANOVA. *n* = 4/group. (**F**) Daily food intake (left) and cumulative food intake for final 5 days of the study (right, starting when animals develop cachexia) in WT and CCR2KO tumor and sham mice. *p<0.05, **p<0.01, ***p<0.001 comparing WT tumor vs. CCR2KO tumor in Bonferroni *post hoc* analysis in two-way ANOVA. *n* = 4/5 per group. Results are representative of three independent experiments. (**G**) Left = mass of dissected gastrocnemius, normalized to initial body weight, at 11 d.p.i. ##p<0.01 for interaction effect between genotype and tumor status in two-way ANOVA analysis. (**H**) qRT-PCR analysis of *Fbxo32*, *Trim63*, and *Foxo1* from RNA extracted from gastrocnemii dissected at 11 d.p.i. Values normalized to those from WT sham. *p<0.05, WT tumor vs. CCR2KO tumor dCt values. *n* = 3–5/group.

The online version of this article includes the following figure supplement(s) for figure 7:

**Figure supplement 1.** The CCR2-CCL2 axis is of selective importance for the brain in PDAC cachexia.

**Figure supplement 2.** RNASeq of neutrophils in different organs during PDAC.

**Figure supplement 3.** Expression of neutrophil 'brain-specific' transcripts.

differentially expressed in the brain-infiltrating neutrophils compared to those in the liver, tumor, and circulation (*Figure 7—figure supplement 2C-E* and *Figure 7—figure supplement 3*).

## Blockade of P2RX7 in the CNS prevents immune cell infiltration into the brain and attenuates cachexia during PDAC

To evaluate the effects of CNS inflammatory responses during PDAC independent of potential systemic effects, we treated mice with intracerebroventricular (ICV) oxidized ATP (oATP). This potently blocks purinergic receptor P2RX7 signaling on brain resident macrophages. Signaling through this receptor is key for neutrophil recruitment to the brain during neuroinflammation (*Roth et al., 2014*). Animals were surgically implanted with indwelling lateral ventricle cannulas, then inoculated IP with KPC cells 1 week later. Mice received daily ICV injections of either 500 ng oATP or vehicle (aCSF), starting 3 d.p.i. (*Figure 8A*). oATP treatment completely prevented both neutrophils and total CD45$^{high}$ myeloid cells from infiltrating the brain (*Figure 8B*). There was a nonsignificant decrease in Ly6C$^{high}$ monocytes in oATP-treated tumor mice (*Figure 8B*). Ly6C$^{low}$ myeloid cells and T-cells were not affected (*Figure 8—figure supplement 1C*). Furthermore, ICV oATP treatment did not affect any circulating immune cell population (*Figure 8—figure supplement 1D*). When we investigated infiltrating immune cells in the VI, both CD45+ globoid cells and CD45+MPO+ neutrophils in the VI were greatly decreased in oATP-treated tumor animals compared to aCSF-treated tumor animals (*Figure 8C and D*). We also observed that oATP treatment attenuated anorexia in tumor mice (*Figure 8E*). There was trend toward increased gastrocnemius mass (p=0.09) in oATP-treated tumor mice compared to aCSF-treated tumor bearing mice (*Figure 8F*), which corresponded to a trend toward decreased induction of genes associated with proteolysis in gastrocnemius muscle (*Figure 8—figure supplement 1A*), demonstrating that muscle catabolism was moderately attenuated by oATP administration directly into the brain. Tumor size in oATP-treated tumor mice was identical to that of aCSF-treated tumor mice (*Figure 8—figure supplement 1B*).

Since ICV oATP antagonizes P2RX7 on brain macrophages, we investigated its effect on microglia. To quantify activation state, we first assessed microglia morphology in the hippocampus. We did not observe any differences in microglia size, Iba-1 staining area, and Iba-1 intensity per cell when comparing aCSF- or oATP-treated tumor animals to oATP-treated sham animals or each other (*Figure 8—figure supplement 2A and B*). We also assessed microglia activation state by using qRT-PCR to quantify expression of transcripts associated with microglia activation in the hippocampus. We observed no differences in expression of *Tnf*, *Cd68*, *Tmem119*, *P2y12*, and *P2x7r* in oATP-treated tumor-bearing animals compared to aCSF-treated tumor-bearing animals (*Figure 8—figure supplement 2C*).

## Discussion

Several lines of investigation show that production of inflammatory mediators in the brain correlates strongly with CNS-mediated symptoms during cancer (*Burfeind et al., 2018*; *Michaelis et al., 2017*), yet the impact of neuroinflammation during malignancy is still not well understood. Our data show that in a mouse model of PDAC, myeloid cells, consisting predominately of neutrophils, infiltrate the brain, in a CCR2-dependent manner, where they drive anorexia and muscle catabolism. We observed that infiltrating immune cells accumulated specifically in a unique layer of meninges called the velum interpositum (VI), which is adjacent to the hippocampus and the habenula, the latter of which is important for appetite regulation and is associating with cachexia in humans (*Maldonado et al., 2018*). We observed robust *Ccl2* mRNA and protein expression, along with CCR2+ neutrophils, exclusively in this region. The VI is implicated as a key structure for initial immune infiltration during states of neuroinflammation such as EAE (*Schmitt et al., 2012*) and traumatic brain injury (*Szmydynger-Chodobska et al., 2016*). Indeed, the VI contains the pial microvessels that are a key aspect of the 'gateway reflex', a neuro-immune pathway that involves interactions between leukocytes and neurons involved in stress response (*Tanaka et al., 2017*) and is implicated in gastrointestinal dysfunction during EAE (*Arima et al., 2017*). While we observed myeloid cell infiltration throughout the VI, we also observed sporadic accumulation of neutrophils and other leukocytes around the same pial vessels involved in the gateway reflex (*Arima et al., 2017*) (data not shown, we were unable to quantify these cells due to the sporadic nature of cell infiltration). The role of the gateway reflex in feeding behavior is unknown. It is possible that, in our model of PDAC,

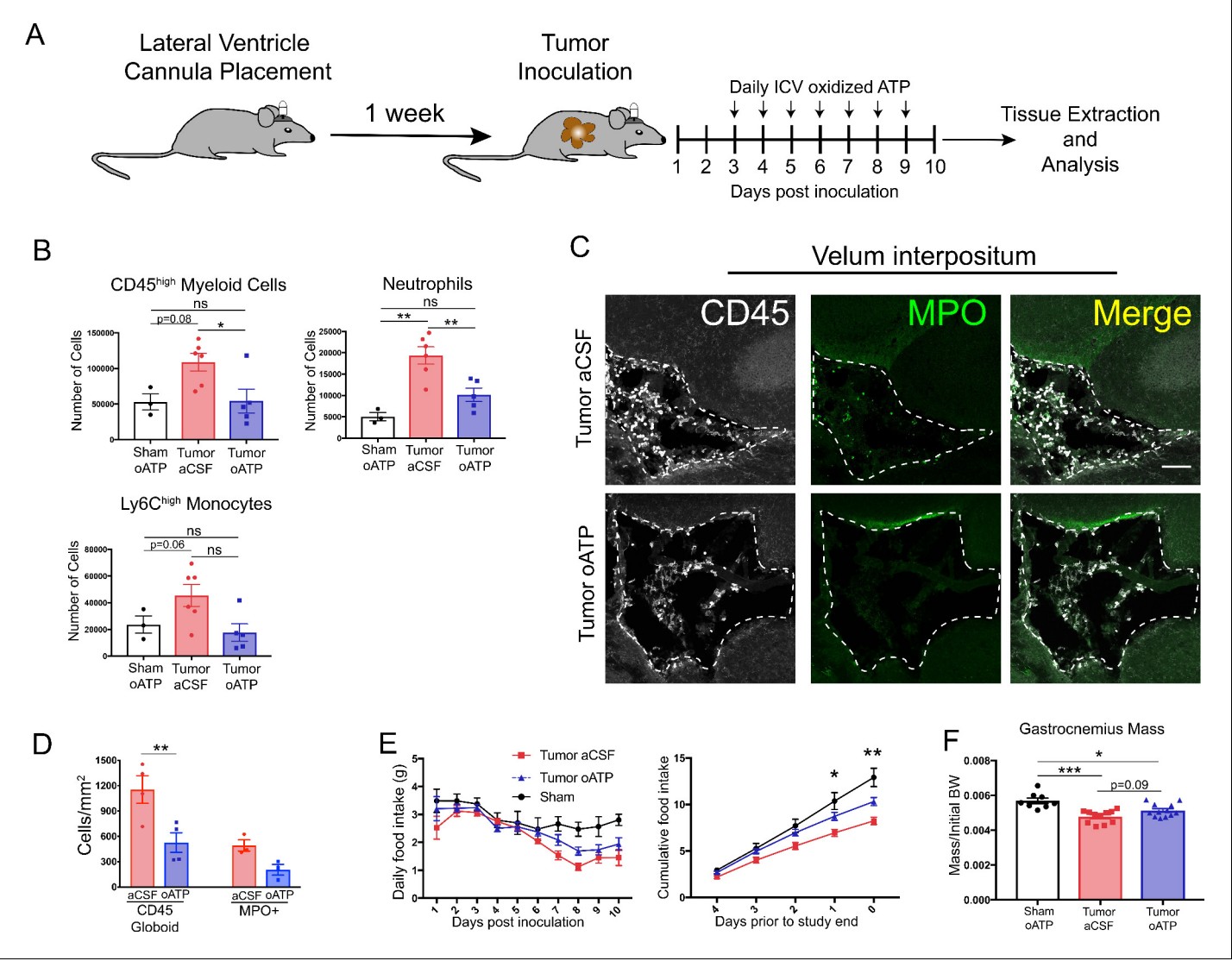

**Figure 8.** Intracerebroventricular administration of oxidized ATP prevents immune cell recruitment to the brain and attenuates anorexia during PDAC. (**A**) Diagram depicting workflow for lateral ventricle cannulation and ICV oATP treatment during PDAC. ICV = intracerebroventricular. (**B**) Quantification of immune cells isolated from whole brain homogenate. *p<0.05, **p<0.01, in Bonferroni *post hoc* analysis in two-way ANOVA. ns = not significant. *n* = 4–7/group. (**C**) Representative 20X confocal microscopy images of the VI from aCSF-treated and oATP-treated tumor animals. Dashed lines denote VI border. Scale bar = 100 µm. (**D**) Quantification of VI-infiltrating CD45+ globoid and MPO+ cells, comparing tumor aCSF to tumor oATP-treated animals. n = 3–4/group, 4 VI images per animal. **p<0.01, in Bonferroni *post hoc* analysis in one-way ANOVA. (**E**) Daily food intake (left) and cumulative food intake for the final 5 days of the study (right, starting when animals develop symptoms) *p<0.05, **p<0.01, comparing aCSF tumor vs. oATP tumor in Bonferroni *post hoc* analysis in two-way ANOVA. *n* = 8–12/group. Results consist of two independent experiments pooled (*n* = 4–7/group in each experiment). (**F**) Mass of dissected gastrocnemius, normalized to initial body weight, at 10 d.p.i. *P < 0.05, **P < 0.01, in Bonferroni post hoc analysis in two-way ANOVA. BW = body weight.

The online version of this article includes the following figure supplement(s) for figure 8:

**Figure supplement 1.** Intracerebroventricular antagonism of P2RX7 does not affect systemic inflammation or tumor size during PDAC.

**Figure supplement 2.** Intracerebroventricular administration of oxidized ATP does not affect microglia activation during PDAC.

brain infiltrating neutrophils were involved in generating anorexia and muscle catabolism via a neuro-immune circuit similar to the gateway reflex, involving inflammation generated in the VI, and possibly transmitted to the habenula, or other regions involved in appetite regulation.

The role and presence of infiltrating leukocytes in the CNS during systemic inflammation remain poorly understood. While previous reports show that neutrophils infiltrate the brain after septic

doses of LPS or sepsis induced by cecal ligation (*He et al., 2016*), it is still unknown if they contribute to neurologic sequelae (anorexia, fatigue, cognition and memory deficits, etc.) during and after sepsis. A series of studies utilizing a mouse model of inflammatory liver disease showed that 'sickness behaviors' could be attenuated if myeloid cell recruitment to the brain was abrogated via any one of several different interventions, including: 1) administration of a P-selectin inhibitor (*Kerfoot et al., 2006*), 2) deleting *Ccr2* (*D'Mello et al., 2009*), and 3) inhibiting microglia activation with minocycline (*D'Mello et al., 2013*). However, unlike our study, these studies did not address many CNS-mediated signs and symptoms associated with chronic disease, including anorexia and muscle catabolism, instead using social interaction as their sole measure of sickness behaviors. They also did not address whether their interventions affected monocyte infiltration in other tissues. Therefore, our results, along with previous studies, implicate brain-infiltrating myeloid cells as key players in driving CNS-mediated signs and symptoms during inflammatory disease.

We were unable to deplete neutrophils, despite daily IP injections of high-dose anti-Ly6G antibody. This dosing regimen was previously thought to thoroughly deplete neutrophils (*Daley et al., 2008*; *Reber et al., 2017*). However, most studies used Ly6G labeling to verify depletion, which can result in incorrectly assuming that neutrophils are depleted. This issue was discussed in a recent study, which demonstrated that despite lack of Ly6G labeling, neutrophils are still present in conditions of immune activation and even increased in some compartments (*Pollenus et al., 2019*). In our PDAC model, this is problematic since there is massive neutrophil mobilization from the bone marrow. Moreover, previous studies demonstrated that PDAC reprograms macrophages, effectively impairing their phagocytic capabilities (*Liu et al., 2019*; *Mantovani and Sica, 2010*), Since macrophage phagocytosis was shown to be required for Ly6G antibody-mediated neutrophil depletion (*Bruhn et al., 2016*), this may explain why the Ly6G antibody bound to neutrophils, but they were not depleted.

We observed a decrease in total number of lymphocytes in the brain starting at 5 d.p.i., which persisted throughout the course of PDAC. This was driven by a decrease in B-cells and CD4+ T-cells. We chose not pursue this further as the purpose of this series of studies was to investigate infiltrating immune cells in the brain. However, with the recent discovery of the meningeal lymphatics and immune surveillance in the brain (*Louveau et al., 2018*), the role of lymphocytes in brain immune regulation is beginning to be appreciated. While the vast majority of lymphocytes in the non-inflamed murine brain are intravascular, even after thorough perfusion of the vasculature (*Mrdjen et al., 2018*), there are lymphocytes in the CSF and meningeal lymphatics. Indeed, a decrease in CD4+ T-cells in the brain during PDAC may reflect a loss of immune regulation in the brain, as they cells have a key cerebroprotective role during neuroinflammation (*Liesz et al., 2009*).

Hypothalamic microglia were previously implicated in orchestrating a multicellular hypothalamic immune response, including infiltrating myeloid cells, during high fat diet-induced obesity (*Valdearcos et al., 2017*). While we did not observe neutrophil infiltration in the hypothalamus during PDAC, we did observe infiltration of non-neutrophil C45+ globoid cells in the meninges surrounding the median eminence. This population of cells should be identified in future studies, and the role of these cells and hypothalamic microglia in PDAC cachexia should be investigated.

We showed that CCR2 inhibition, but not CXCR2 inhibition, attenuated immune cell infiltration into the brain and resulted in decreased anorexia during PDAC. *Ccr2* deletion resulted in even greater attenuation in cachexia and immune cell infiltration into the brain. The difference between genetic and pharmacologic CCR2 blockade is likely due to incomplete blockade of CCR2 signaling with RS504393. It should be noted that both RS504393 and SB225002 treatment resulted in a non-significant decrease in neutrophils in the VI, yet only RS504393 treatment decreased cachexia, suggesting that monocytes (since CCR2 is typically important in monocyte chemotaxis) may be important mediators of cachexia. Also, it is worth noting that our pharmacologic inhibition of CXCR2 may have been incomplete. Therefore, it remains to be determined whether CXCR2 ligands are important for recruitment of cachexia-inducing neutrophils to the CNS. Regardless, our results are in agreement with previous studies investigating sickness behaviors during inflammatory liver disease, which showed that CCR2KO mice exhibited attenuated monocyte infiltration into the brain, along with decreased sickness behaviors (*D'Mello et al., 2009*). Furthermore, it was recently reported that mice lacking *Ccr2* had decreased myeloid cell infiltration into the brain and attenuated cognitive impairment during a model of sepsis induced by *Streptococcus pneumoniae* injection into the lungs (*Andonegui et al., 2018*). In an attempt to identify inflammatory biomarkers for PDAC-associated

cachexia, Talbert et al. identified CCL2 as the only cytokine or chemokine (out of a panel of 25) that was increased in the serum of cachectic PDAC patients but not increased in the serum of non-cachectic patients (*Talbert et al., 2018*). It is possible that the differences we observed in gastrocnemius catabolism between WT and CCR2KO tumor animals were due to differences in food intake, but the fact that we observed a significant decrease in induction of the catabolic genes *Mafbx*, *Murf1*, and *Foxo1* in CCR2KO tumor animals, which are not induced by decreased food intake/starvation (*Braun et al., 2011*), makes this unlikely. However, CCR2KO tumor-bearing mice still experienced anorexia and muscle catabolism compared to CCR2KO sham mice, reflecting incomplete resolution of cachexia. It remains possible that more prominent factors, yet to be identified, contribute to cachexia during PDAC.

While CCR2 is usually not considered a key receptor for neutrophil recruitment, previous studies show it is important for neutrophil chemotaxis during sepsis (*Souto et al., 2009*; *Souto et al., 2011*). Interestingly, while we observed a robust decrease in brain-infiltrating neutrophils in CCR2KO mice, we did not observe a decrease in liver- or tumor-infiltrating neutrophils, indicating that CCR2 is important for neutrophil recruitment specifically to the brain. Circulating neutrophils in sham animals did not express CCR2, but a small percentage of circulating neutrophils expressed CCR2 in tumor animals, suggesting that the presence of a tumor induces CCR2 expression in neutrophils. Furthermore, a significant percentage of neutrophils in the brain expressed CCR2 during PDAC, meaning a distinct population of neutrophils is recruited to the brain from the circulation. These results, suggest that the population recruited to the brain has a distinct function from those recruited to other organs.

We administered oxidized ATP, a purinergic receptor antagonist, directly into the brain and observed complete abrogation of circulating myeloid cell recruitment to the brain in tumor animals, as well as anorexia attenuation. These results show that brain inflammation is an important driver of PDAC-associated anorexia. While there was no change in microglia morphology after oATP administration, consistent with previous studies (*Martin et al., 2019*; *Roth et al., 2014*), nor was there a change in expression of genes associated with microglia activation, we cannot rule out the possibility that the difference in anorexia we observed were due to changes in microglia phenotype. The presence of an indwelling lateral ventricle cannula may have also induced microglia activation and influenced morphology quantification. However, we did take care to acquire images from the contralateral hemisphere. It is also worth noting that, unlike our previous experiments in non-cannulated mice, we did not observe a significant increase in brain-infiltrating Ly6C$^{high}$ monocytes in aCSF-treated tumor mice compared to oATP-treated sham mice. However, this comparison was nearly significant (p=0.06), and the lack of significance was likely due to heterogeneous results caused by cannulation. Furthermore, we observed an increased Ly6C$^{high}$ monocyte infiltrate in our aCSF-treated tumor animals compared to non-cannulated tumor animals, suggesting the indwelling lateral ventricle cannula did affect the inflammatory response in the brain to at least a small degree. Nevertheless, oATP completely prevented myeloid cells from infiltrating the brain during PDAC, strongly implicating these cells as mediators of anorexia.

We observed that the transcriptional profile of brain-infiltrating neutrophils was distinct from those in the circulation, liver, and tumor. It was previously demonstrated that the CNS induces a more 'inflammatory' transcriptional profile in infiltrating myeloid cells compared to those that infiltrate other organs (*Spath et al., 2017*). Reasons for this are not entirely clear, but may be due to relative lack of regulatory T-cells, the blood brain barrier preventing access to soluble anti-inflammatory factors, and presence of immunogenic substances more abundant in the brain, such as myelin-associated lipids.

A few limitations should be considered when interpreting results of this study. First, our data were produced in a single model of pancreatic cancer. While our model is extensively characterized and reliably recapitulates many of the CNS-mediated symptoms observed in humans, other malignancies should also be considered. Second, it is possible, even likely, that circulating immune cells infiltrate and influence function in other organs dysfunctional during cancer (skeletal muscle, adipose tissue, etc.). However, the purpose of this study was to investigate and characterize interactions between circulating immune cells and the brain during PDAC. Therefore, we chose to focus specifically on the brain so as to not overcomplicate analysis. Third, since we were not able to selectively deplete neutrophils, we cannot definitely conclude that these cells are the sole cellular drivers of cachexia during PDAC. We observed a small increase in Ly6C$^{hi}$ monocytes in the brains of animals

during PDAC, which was also attenuated by CCR2 deletion. Therefore, these cells may have contributed to anorexia and muscle catabolism. However, there were far fewer Ly6C$^{hi}$ monocytes ($\approx$2,000) in the brain than neutrophils ($\approx$9,000) during PDAC, and these cells only constituted about 15% of brain CD45$^{high}$ myeloid cells (vs. approximately 50% for neutrophils). Fourth, it is possible that the tumor metastasized to the brain, which drove the immune response. However, our lab has conducted countless PDAC experiments in mice and analyzed thousands of brain slices. We have yet to observe a single metastasis. Lastly, despite our extensive analysis, we cannot rule out with absolute certainty that the differences we observed in RS504393-treated or CCR2KO mice were not due to differences in tumor response. However, both the CCR2/CCL2 axis and neutrophils are reported to be 'pro-tumor' (*Coffelt et al., 2016*; *Qian et al., 2011*) and therefore systemic treatment targeting neutrophils or the CCR2/CCL2 axis in humans may be particularly beneficial in that they decrease tumor size and abrogate CNS dysfunction. This would be advantageous to conventional anti-tumor therapies such as chemotherapy and checkpoint inhibitors, which are both known to cause cachexia-like symptoms (*Braun et al., 2014*; *Michot et al., 2016*) and not effective against PDAC.

In summary, we demonstrated that myeloid cells infiltrate the CNS throughout the course of PDAC and that preventing myeloid cells from infiltrating the brain attenuates anorexia and muscle catabolism. We showed there are distinct mechanisms for immune cell recruitment to the brain during systemic inflammation, and demonstrated a novel role for CCR2 in neutrophil recruitment to the brain, providing key insights into mechanisms of neuroinflammation and associated symptoms.

## Materials and methods

### Animals and treatment

Male and female 20–25 g WT C57BL/6J (stock no. 000664), *Ptprc$^a$*-EGFP (aka Ly5.1-eGFP, stock no. 002014), *Ccl2*$^{mCherry}$ (stock no. 016849), *Ccr2*$^{RFP}$ (stock no. 017586), and CCR2KO (stock no. 004999) were purchased at Jackson Laboratories. Animals were aged between 7 and 12 weeks at the time of study and maintained at 27°C on a normal 12:12 hr light/dark cycle and provided ad libitum access to water and food. Experiments were conducted in accordance with the National Institutes of Health Guide for the Care and Use of Laboratory Animals, and approved by the Animal Care and Use Committee of Oregon Health and Science University.

RS504393 was dissolved in DMSO and SB225002 was dissolved in ethanol. 5 mg/kg RS504393 or 10 mg/kg SB225002 were injected subcutaneously twice daily starting at 3 d.p.i. in PDAC-bearing mice. Vehicle-treated animals received the same volume in DMSO injected subcutaneously twice daily.

For neutrophil depletion experiments, 500 μg of anti-Ly6G antibody (clone 1A8, Biolegend) or isotype control was administered IP daily in 200 μL PBS, starting at 2 d.p.i.

### KPC cancer model

Our lab generated a mouse model of PDAC initiated by a single IP or orthotopic injection of 1-5e$^6$ murine-derived KPC PDAC cells (*Michaelis et al., 2017*). These cells are derived from tumors in C57BL/6 mice heterozygous for oncogenic *Kras*$^{G12D}$ and point mutant T$\underline{p}$53$^{R172H}$ with expression targeted to the pancreas via the *P*dx1-$\underline{C}$re driver (*Foley et al., 2015*). Cells were maintained in RPMI supplemented with 10% heat-inactivated FBS, and 50 U/mL penicillin/streptomycin (Gibco, Thermofisher), in incubators maintained at 37°C and 5% $CO_2$. In the week prior to tumor implantation, animals were transitioned to individual housing to acclimate to experimental conditions. Animal food intake and body weight were measured once daily. Sham-operated animals received PBS in the same volume. Bedding was sifted daily to account for food spillage not captured by cagetop food intake measurement. Animals were euthanized between 9 and 14 days post-inoculation, when food intake was consistently decreased and locomotor activity was visibly reduced, yet signs of end-stage disease (ascites, unkempt fur, hypotheremia, etc.) were not present (*Michaelis et al., 2017*). We extensively characterized this model and demonstrated that IP and OT inoculation resulted in similar cachexia progression and inflammatory response (both in the CNS and systemically) (*Burfeind et al., 2018*; *Michaelis et al., 2017*; *Zhu et al., 2019*).

## Generation of Ly5.1-EGFP chimera mice

WT C57BL/6J male mice aged 8–10 weeks were injected IP with the alkylating agent treosulfan (Ovastat, a generous gift from Joachim Baumgart at Medac GmbH, Germany) at a dose of 1500 mg/kg/day for three consecutive days prior to the day of bone marrow transplant (BMT). 24 hr after the third treosulfan injection, a Ly5.1-EGFP male or female donor mouse aged between 2 and 6 months was euthanized and femurs, tibias, humeri, and radii were dissected. After muscle and connective tissue were removed, marrow cells were harvested by flushing the marrow cavity of dissected bones using a 25-gauge needle with Iscove's modified Dulbecco's medium supplemented with 10% FBS. The harvested cells were treated with RBC lysis buffer, filtered with a 70 µm cell strainer, and counted. $3–4 \times 10^6$ cells in 200 µL HBSS were transplanted immediately into each recipient mouse via tail vein injection. To prevent infection during an immunocompromised period, recipient mice received amoxicillin dissolved in their drinking water (150 mg/L) for 2 weeks starting on the first day of treosulfan injection. GFP BMT mice were given at least 5 weeks for marrow reconstitution and recovery. Percent chimerism in each GFP BMT mouse was determined by flow cytometry analysis of circulating leukocytes.

## Intracerebroventricular cannulation and injections

Mice were anesthetized under isoflurane and placed on a stereotactic alignment instrument (Kopf Instruments). 26-gauge lateral ventricle cannulas were placed at 1.0 mm X, −0.5 mm Y, and −2.25 mm Z relative to bregma. Mice were given one week for recovery after cannula placement. Injections were given in 2 µl total volume. Oxidized ATP was dissolved in aCSF and injected at a concentration of 250 ng/µL over 5 min while mice were anesthetized under isoflurane.

## Immunofluorescence immunohistochemistry

Mice were anesthetized using a ketamine/xylazine/acetapromide cocktail and sacrificed by transcardial perfusion fixation with 15 mL ice cold 0.01 M PBS followed by 25 mL 4% paraformaldehyde (PFA) in 0.01 M PBS. Brains were post-fixed in 4% PFA overnight at 4℃ and cryoprotected in 20% sucrose for 24 hr at 4℃ before being stored at −80℃ until used for immunohistochemistry. Immunofluorescence immunohistochemistry was performed as described below. Free-floating sections were cut at 30 µm from perfused brains using a Leica sliding microtome. Sections were incubated for 30 min at room temperature in blocking reagent (5% normal donkey serum in 0.01 M PBS and 0.3% Triton X-100). After the initial blocking step, sections were incubated in primary antibody (listed below) in blocking reagent for 24 hr at 4℃, followed by incubation in secondary antibody (also listed below) for 2 hr at room temperature. Between each stage, sections were washed thoroughly with 0.01 M PBS. Sections were mounted onto gelatin-coated slides and coverslipped using Prolong Gold antifade media with DAPI (Thermofisher).

The following primary anti-mouse antibodies were used, with company, clone, host species, and concentration indicated in parentheses: CD11b (eBioscience, rat, M1/70, 1:1000), CD45 (BD, rat, 30-F11, 1:1000), myeloperoxidase (R and D, goat, polyclonal, 1:1000), Ly6G (Biolegend, 1A8, rat, 1:250), Iba-1 (Wako, Rabbit, NCNP24, 1:1000), CD206 (Bio-rad, rat, MR5D3, 1:1000), ER-TR7 (Abcam, rat, ER-TR7, 1:1000), and citrillunated histone H3 (Abcam, rat, polyclonal, 1:1000). We also used a chicken anti-mCherry antibody (Novus Biologicals, polyclonal, 1:20,000), to amplify mCherry signal in sections from CCL2$^{fl/fl}$ mice and a rabbit anti-RFP antibody (Abcam, polyclonal, 1:1000) to amplify RFP signal in sections from CCR2$^{RFP/WT}$ mice.

The following secondary antibodies were used, all derived from donkey and purchased from Invitrogen, with dilution in parentheses: anti-goat AF488 (1:500), anti-rabbit AF555 (1:1000), anti-rat AF555 (1:1000), anti-rat AF633 (1:500), and anti-chicken AF555 (1:1000).

## Image acquisition and analysis

All images were acquired using a Nikon confocal microscope. Cell quantification was performed on 20X images using the Fiji Cell Counter plugin by a blinded researcher. CD45+ cells were defined as CD45 bright globoid cells, and neutrophils were defined as CD45+ MPO+ cells. The velum interpositum (VI) was defined as the layer of meninges (identified by appearance of staining background) between the hippocampus and thalamus, from bregma −1.7 to −2.6 mm. At least 8 VI images were quantified from each animal. The median eminence was defined as the base of the mediobasal

hypothalamus (far ventral part of the brain), adjacent to the third ventricle from bregma −1.95 to −2.5 mm. Four ME images were quantified from each animal. The area postrema was defined as the region in from bregma −7.2 to −7.75 mm. Four area postrema images were quantified from each animal.

## Microglia morphology analysis

Microglia activation in the hippocampus was quantified using Fiji (ImageJ, NIH). Five images of the dentate gyrus were acquired from each animal. Images were 2048 x 2048 pixels, with a pixel size of 0.315 µm. Images were uploaded to Fiji by a blinded reviewer (KGB) and converted to 8-bit grey-scale images. After thresholding, microglia were identified using the 'analyze particle' function, which measured mean Iba-1 fluorescent intensity per cell, cell area, and percent area covered by Iba-1 staining.

## Primary microglia culture and PDAC-conditioned media treatment

Primary mixed-glial cultures containing microglia and astrocytes were prepared from neonatal mouse cortices. Brain cortices from 1- to 3-day-old newborn mouse pups were dissected, then digested with papain (Worthington Biochemical Corporation). Cells were passed through a 70 µm cell strainer and seeded in 75 cm$^2$ flasks in DMEM media (low glucose with L-glutamine, 10% FBS and 1% penicillin/streptomycin). Microglia were isolated 14–16 days later by shaking flasks at 200 rpm at 37℃ for 1 hr. Cells were re-plated into six well plates at $5 \times 10^5$/well and maintained in DMEM media for 24 hr before stimulation. More than 90% of these isolated cells were confirmed as microglia by Iba1 staining and flow cytometry (CD45+CD11b+ cells, data not shown).

PDAC tumor cells were cultured in a 75 cm$^2$ flask until confluent. 24 hr prior to treatment, 13 ml fresh media (RPMI supplemented with 10% FBS and 1% penicillin-streptomycin) was added for generating PDAC-conditioned media. On the treatment day, 4 mL PDAC-conditioned media mixed with 1 mL fresh RPMI media (to ensure treated microglia were not nutrient starved) was added to each of the three wells of the six well plate containing microglia. The other three wells each received 5 mL control media (RPMI media). Three additional wells received 5 mL control media containing 10 ng LPS. 16 hr after treatment, media was removed, and adherent cells were washed with PBS then lysed. RNA was then extracted from cell lysate using a Qiagen RNAEasy kit.

## In situ hybridization

At 10 d.p.i., mice were euthanized with CO$_2$ and brains were removed then frozen on dry ice. 20 µm coronal sections were cut on a cryostat and thaw-mounted onto Superfrost Plus slides (VWR Scientific). Sections were collected in a 1:6 series from the diagonal band of Broca (bregma 0.50 mm) caudally through the mammillary bodies (bregma 5.00 mm). 0.15 pmol/ml of an antisense $^{33}$P-labeled mouse *Ccl2* riboprobe (corresponding to bases 38–447 of mouse *Ccl2*; GenBank accession no. NM_011333.3) was denatured, dissolved in hybridization buffer along with 1.7 mg/ ml tRNA, and applied to slides. Slides were covered with glass coverslips, placed in a humid chamber, and incubated overnight at 55℃. The following day, slides were treated with RNase A and washed under conditions of increasing stringency. Slides were dipped in 100% ethanol, air dried, and then dipped in NTB-2 liquid emulsion (Kodak). Slides were developed 4 d later and cover slipped.

## Quantitative Real-Time PCR

Prior to tissue extraction, mice were euthanized with a lethal dose of a ketamine/xylazine/acetapromide and sacrificed. Hippocampal blocks and gastrocnemii were dissected, snap frozen, and stored in −80℃ until analysis. RNA was extracted using an RNeasy mini kit (Qiagen) according to the manufacturer's instructions. cDNA was transcribed using TaqMan reverse transcription reagents and random hexamers according to the manufacturer's instructions. PCR reactions were run on an ABI 7300 (Applied Biosystems), using TaqMan universal PCR master mix with the following TaqMan mouse gene expression assays: *18* s (Mm04277571_s1), *Tnf* (Mm00443258_m1), *Il6* (Mm01210732_g1), *Il-1β* (Mm00434228_m1), *Il10* (Mm01288386_m1), *Ccl2* (Mm99999056_m1), *Ccl3* (Mm00441259_g1), *Ccl5* (Mm01302427_m1), *Cxcl1* (Mm04207460_m1), *Cxcl2* (Mm00436450_m1), *Cxcl5* (Mm00436451_g1), *Cxcl9* (Mm00434946_m1), *Cxcl10* (Mm00445235_m1), *Cd68* (Mm03047343_m1), *Tmem119* (Mm0052305_m1), *P2y12* (Mm01950543_S1), *P2r* × *7* (Mm00446026_m1), *Gapdh*

(Mm99999915_g1), *Fbxo32* (Mm00499518_m1), *Trim63* (Mm01185221_m1), and *Foxo1* (Mm00490672_m1).

Relative expression was calculated using the ΔΔCt method and normalized to WT vehicle treated or sham control. Statistical analysis was performed on the normally distributed ΔCt values.

## Hematology

Whole blood was analyzed with a veterinary hematology analyser (HemaVet, 950FS, Drew Scientific, Oxford, CT) to assess total white blood cell count and white blood cell differential.

## Flow cytometry

Mice were anesthetized using a ketamine/xylazine/acetapromide cocktail and perfused with 15 mL ice cold 0.01 M PBS to remove circulating leukocytes. If circulating leukocytes were analyzed, blood was drawn prior to perfusion via cardiac puncture using a 25-gauge needle, then placed in an EDTA coated tube. After perfusion, organs were extracted and immune cells were isolated using the following protocols:

### Brain

Brains were minced in a digestion solution containing 1 mg/mL type II collagenase (Sigma) and 1% DNAse (Sigma) in RPMI, then placed in a 37°C incubator for 45 min. After digestion, myelin was removed via using 30% percoll in RPMI. Isolated cells were washed with RPMI, incubated in Fc block for 5 min, then incubated in 100 μL of PBS containing antibodies for 30 min at 4°C. Cells were then washed once with RPMI.

### Liver

Livers were pushed through a 70 μm nylon strainer, then washed once with RPMI. The resulting suspension was resuspended in a 40 mL digestion solution containing 1 mg/mL type II collagenase (Sigma) and 1% DNAse (Sigma) in RPMI, then placed in a 37°C incubator for 1 hr. After digestion, the suspension was placed on ice for 5 min, then the top 35 mL was discarded. The remaining 5 mL was washed in RPMI, resuspended in 10 mL 35% percoll to remove debris, then treated with RBC lysis buffer. The resulting cell suspension was washed with RPMI, then cells were incubated in 100 μL of PBS containing antibodies for 30 min, then washed with RPMI.

### Tumor

A 0.4–0.5 g piece of pancreatic tumor was removed, minced in a digestion solution containing 1 mg/mL type II collagenase (Sigma) and 1% DNAse (Sigma) in RPMI, then placed in a 37°C incubator for 1 hr. After digestion, cells were washed with RPMI, then incubated in 100 μL of PBS containing antibodies for 30 min at 4°C. Cells were then washed once with RPMI.

### Blood

200 μL of blood was drawn via cardiac puncture with a 25-gauge needle. Red blood cells were then lysed with 1X RBC lysis buffer. The resulting cell suspension was washed with RPMI, then cells were incubated in 100 μL of PBS containing antibodies for 30 min at 4°C, then washed with RPMI.

### Gating Strategy

Cells were gated on LD, SSC singlet, and FSC singlet. Immune cells were defined as CD45+ cells. In the brain, microglia were defined as CD45$^{mid}$CD11b+. Leukocytes were identified as either myeloid cells (CD45$^{high}$CD11b+ in the brain, CD45+CD11b+ in all other tissues) or lymphocytes (CD45$^{high}$CD11b- in the brain, CD45+CD11b- in all other tissues). From myeloid cells Ly6C$^{low}$ monocytes (Ly6C$^{low}$Ly6G-), Ly6C$^{high}$ monocytes (Ly6C$^{high}$Ly6G-), and neutrophils (Ly6C$^{mid}$Ly6G+) were identified. From lymphocytes, CD3+ cells were identified as T-cells, and further phenotyped as CD4 + or CD8+ T-cells. CD3- T-cells were divided into NK1.1+ NK cells or CD19+ B-cells. Flow cytometry analysis was performed on a BD Fortessa, Symphony, or LSRII analytic flow cytometer.

## Antibodies

All antibodies were purchased from BioLegend, except for Live/Dead, which was purchased from Invitrogen (Fixable Aqua, used at 1:200 dilution). The following anti-mouse antibodies were used, with clone, fluorophore, and dilution indicated in parenthesis: CD3 (17A2, PE, 1:100), CD3 (17A2, APC/Cy7, 1:400), CD4 (RM4-5, APC, 1:100), CD8 (53–6.7, APC/Cy7, 1:800), CD11b (M1/70, APC, 1:800), CD11b (M1/70, FITC, 1:200), CD19 (6D5,BV650, 1:33), CD45 (30-F11, PerCP/Cy5.5, 1:400), CD45 (30-F11, APC/Cy7, 1:400), CX3CR1 (SAO11F11, PE/Cy7, 1:100), Dec205 (NLDC-145, APC,1:100), CD115 (AFS98, APC/CY7, 1:400), Ly6C (HK1.4, PerCP, 1:100), Ly6G (1A8, PE/Cy7, 1:800), NK1.1 (PK136, BV785, 1:800), CCR2 (SA203G11, AF647, 1:100).

## FACS sorting for RNAseq

At 10 d.p.i., mice were anesthetized and 200 μL of blood was drawn via cardiac puncture with a 25-gauge needle. Circulating leukocytes were then removed via transcardiac perfusion with PBS and brain, liver, and tumor were removed. Leukocytes were isolated from blood, brain, liver, and tumor as described above. Sorting was performed using an Influx sorter (BD) with a 100 μm nozzle. Neutrophils were defined as CD11b$^{high}$Ly6G$^{high}$ live, singlet cells, and were sorted into lysis buffer (Qiagen), then stored at −80° C. Using this technique, our lab regularly achieves 95%+ post-sort purity.

# RNA isolation and sequencing

## RNA Isolation, sequencing, and library preparation

Total RNA was isolated from FACS-sorted CD11b$^{high}$Ly6G$^{high}$ neutrophils using an RNAeasy Plus Micro kit (Qiagen). RNA integrity was verified by a Bioanalyzer (Agilent). Sample cDNAs were prepared using the SMART-Seq v4 Ultra Low Input kit (Takara) using 250 pg of input total RNA followed by library preparation using a TruSeq DNA Nano kit (Illumina). Libraries were verified by Tapestation (Agilent). Library concentrations were determined by real-time PCR with a StepOnePlus Real Time PCR System (Thermo Fisher) and a Kapa Library Quantification Kit (Kapa Biosystems/Roche). Libraries were sequenced with a 100 cycle single read protocol on a HiSeq 2500 (Illumina) with four libraries per lane. Fastq files were assembled using Bcl2Fastq2 (Illumina).

## RNA-seq processing and analysis

Quality control checks were done using the FastQC package (https://www.bioinformatics.babraham.ac.uk/projects/fastqc/). Raw reads were normalized and analyzed using the Bioconductor package DESeq2 (*Love et al., 2014*), which uses negative binomial generalized linear models. Only those genes that were expressed in at least one sample were included in differential expression analysis. To identify transcripts differentially expressed in brain-infiltrating neutrophils compared to neutrophils infiltrating other organs, gene expression in neutrophils isolated from brain was compared to that in neutrophils isolated from liver, tumor, and blood. In order to control for the effects of tumor on circulating neutrophils, genes that also were differentially expressed in circulating neutrophils from tumor mice compared to circulating neutrophils from sham mice were excluded from analysis. All p-values were adjusted for multiple comparisons using the Benjamani-Hochberg method (*Benjamini and Hochberg, 1995*). Differential expression was defined based on statistical significance (adjusted p-value<0.05) and effect size (log$_2$ fold change)$\leq$or $\geq$ −2. Heatmaps were created using the pheatmap package from R. Gene Ontology analysis was performed using the Goseq Bioconductor R package (*Young et al., 2010*). For pathway enrichment analysis, pathway annotation from the Reactome knowledgebase (*Croft et al., 2014*; *Fabregat et al., 2018*) was used.

# Statistical analysis

Data are expressed as means ± SEM. Statistical analysis was performed with Prism 7.0 software (Graphpad Software Corp). When two groups were compared, data were analyzed with either student's t-test or Mann-Whitney U test. When more than two groups were compared, data were analyzed with either one-way (when multiple groups were compared to a single sham group) or two-way (when there were multiple genotypes within tumor and sham groups being compared) ANOVA analysis. For single time point experiments, the two factors in ANOVA analysis were genotype or treatment. In repeated measures experiments, the two factors were group and time. Main effects of genotype, treatment, group, and/or time were first analyzed, and if one effect was significant,

Bonferroni *post hoc* analysis was then performed. For all analyses, significance was assigned at the level of $p < 0.05$.

## Acknowledgements

We thank Pamela Canaday, Dorian LaTocha, and Sara Christensen at the OHSU Flow Cytometry Core for their assistance with FACS sorting for RNASeq; Taylor McFarland at the OHSU Gene Profiling Shared Resource for his assistance with RNA isolation; Amy Carlos and Dr. Robert Searles at the OHSU Massively Parallel Sequencing Core for their assistance with RNA sequencing and library preparation; Dr. Elizabeth Jaffee from Johns Hopkins for providing KPC tumor cells; Drs. Joachim Baumgart and Daniela Reese at Medac GmbH, Germany for providing treosulfan; the Brenden-Colson Center for Pancreatic Care at OHSU for providing funding; and Drs. David Jacoby (OHSU) and Vickie Baracos (University of Alberta) for their help with editing the manuscript. Bioinformatics analytical expertise for this project was supported by partnership between the office of the OHSU Senior Vice President for Research, the University Shared Resources program, the OCTRI Translational Bioinformatics Program (*NIH/NCATS CTSA UL1TR002369*), and the Integrated Genomics Laboratory's Massively Parallel Sequencing Shared Resources. This work was supported by National Institutes of Health grants R01CA184324-01 and R01CA217989-01 to DL Marks, and 5F30CA213745 to KG Burfeind.

## Additional information

### Funding

| Funder | Grant reference number | Author |
| --- | --- | --- |
| National Cancer Institute | R01CA184324-01 | Daniel L Marks |
| National Cancer Institute | R01CA217989-01 | Daniel L Marks |
| Oregon Health and Science University | Brenden-Colson Center for Pancreatic Care | Daniel L Marks |
| National Cancer Institute | 5F30CA213745 | Kevin Glenn Burfeind |

The funders had no role in study design, data collection and interpretation, or the decision to submit the work for publication.

### Author contributions

Kevin G Burfeind, Conceptualization, Data curation, Formal analysis, Methodology, Writing - original draft, Writing - review and editing; Xinxia Zhu, Data curation, Writing - review and editing; Mason A Norgard, Peter R Levasseur, Christian Huisman, Abigail C Buenafe, Data curation; Brennan Olson, Conceptualization, Data curation, Methodology, Writing - review and editing; Katherine A Michaelis, Conceptualization, Methodology, Writing - review and editing; Eileen RS Torres, Jacob Raber, Conceptualization, Writing - review and editing; Sophia Jeng, Formal analysis, Writing - review and editing; Shannon McWeeney, Conceptualization, Formal analysis, Writing - review and editing; Daniel L Marks, Conceptualization, Supervision, Funding acquisition, Project administration, Writing - review and editing

### Author ORCIDs

Kevin G Burfeind https://orcid.org/0000-0002-4192-6753
Katherine A Michaelis http://orcid.org/0000-0002-3225-3649
Eileen RS Torres https://orcid.org/0000-0002-5340-8734
Jacob Raber http://orcid.org/0000-0002-9861-9893
Daniel L Marks https://orcid.org/0000-0003-2675-7047

### Ethics

Animal experimentation: This study was performed in strict accordance with the recommendations in the Guide for the Care and Use of Laboratory Animals of the National Institutes of Health. All of

the animals were handled according to approved institutional animal care and use committee (IACUC) protocols of Oregon Health & Science University. The protocol was approved by the Department of Comparative Medicine of Oregon Health & Science University (protocol IP00038). All surgery was performed under isofluorane anesthesia, and every effort was made to minimize suffering.

## Decision letter and Author response
Decision letter https://doi.org/10.7554/eLife.54095.sa1
Author response https://doi.org/10.7554/eLife.54095.sa2

## Additional files
### Supplementary files
• Transparent reporting form

### Data availability
All data generated or analysed during this study are included in the manuscript and supporting files. Sequencing data is available at GEO (GSE150061).

The following dataset was generated:

| Author(s) | Year | Dataset title | Dataset URL | Database and Identifier |
|---|---|---|---|---|
| Burfeind KG, Jeng S | 2020 | The transcriptional profile of neutrophils in different organs during pancreatic cancer | https://www.ncbi.nlm.nih.gov/geo/query/acc.cgi?acc=GSE150061 | NCBI Gene Expression Omnibus, GSE15006 |

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
