## [Decision Letter]

**Acceptance summary:**

The authors have demonstrated that the progression of pancreatic ductal adenocarcinoma is accompanied by the accumulation of myeloid cells (particularly neutrophils) in the velum interpositum layer of meninges in the brain, which cause neuroinflammation manifested as cancer-related weight loss and anorexia. This work provides a new possibility showing the crosstalk between cancer and the neuro-immune system.

**Decision letter after peer review:**

[Editors’ note: the authors submitted for reconsideration following the decision after peer review. What follows is the decision letter after the first round of review.]

Thank you for submitting your work entitled "A distinct neutrophil population invades the central nervous system during pancreatic cancer" for consideration by *eLife*. Your article has been reviewed by two peer reviewers, one of whom is a member of our Board of Reviewing Editors, and the evaluation has been overseen by a Senior Editor. The following individual involved in review of your submission has agreed to reveal their identity: Alban Gaultier (Reviewer #2).

Our decision has been reached after consultation between the reviewers. Based on these discussions and the individual reviews below, we regret to inform you that your work will not be considered further for publication in *eLife*.

In the current manuscript, Burfeind et al. present an intriguing concept that neutrophil infiltration of the central nervous system (CNS) can play a role in peripheral phenotypes (e.g. anorexia and altered muscle catabolism) associated with tumor progression in a mouse model of pancreatic ductal adenocarcinoma (PDAC). Flow cytometry, bone marrow chimeras and immunofluorescence were used to establish CNS neutrophil infiltration during a time when significant wasting and anorexia are observed (i.e. 10 days post inoculation (d.p.i.)), with one of the sites being within the velum interpositum (VI) meningeal layer that surrounds the hippocampus. Localization of these cells near the hippocampus provided the impetus to assess behavioral and cognitive function using open field (OFA) and novel object recognition (NOR) assays, respectively. Although NOR still demonstrated preferences for exploration of a novel object, suggesting cognitive function remained undisturbed, data using their paradigm of OFA presented with alterations that were interpreted as increased levels of anxiety in tumor bearing animals. Delving further into mechanism by which neutrophils may be recruited to the CNS, CCL2 expression in the VI was suggested to recruit CCR2+ neutrophils. RNAseq of flow cytometrically sorted neutrophils was conducted to demonstrate that these neutrophils represented a unique subset that were transcriptionally distinct from peripheral neutrophils. Furthermore, CCR2 KO animals were demonstrated to have a significant attenuation of anorexia and decreased muscle loss. Finally, intracerebroventricular (ICV) injection of oxidized ATP (oATP), which blocks P2RX7 signaling, resulted in decreased recruitment of neutrophils in the VI and partial amelioration of anorexia, with no significant alterations in muscle loss. Overall this manuscript attempts to like recruitment neutrophils to the CNS during tumor progression that then acts to elicit alterations in anxiety, wasting and anorexia.

Overall this manuscript provides an interesting concept and is well written. The data are interesting and could give insight into the interplay between neuroinflammation and peripheral outcomes during tumor progression but there are issues that need to be addressed. As we judge it will likely take longer than two months to attend to these points, we are returning your paper to you now in case you wish to submit it elsewhere. However, should you wish to resubmit to *eLife*, we would consider a new submission of your manuscript that addresses the points laid out in the reviewer comments below.

1) In order to provide convincing evidence that neutrophils are significantly increased in the CNS, additional flow cytometric data will need to be conducted. As it stands, the gating for CD45^high^CD11b+ cells may still include microglia, which may also non-specifically bind to antibodies and thus look "positive" for markers (e.g. Ly6G+). Although it is appreciated that the authors attempt to use a bone marrow chimera using chemotherapy based myeloablation (i.e. treosulfan/Ovastat) to circumvent blood brain barrier (BBB) disruptions, the gates in which the GFP+ cells fall in Figure 1—figure supplement 2B could still encompass microglia. Since this is a pivotal point for the paper, it would be crucial to re-conduct some of these experiments utilizing a marker which will eliminate microglia (e.g. *Tmem119*) to rule out this caveat. This impacts the entire paper since many of the results are based on how they gated for neutrophils in general, identify CCR2+ cells as including a subset of neutrophils in their flow cytometric studies, and sorting for RNAseq is theoretically also utilizing a similar gate.

2) Although the authors identify a behavioral phenotype using open field assay (OFA), a two-day test for OFA is highly atypical. There is no training day for OFA, and the results found on day 1 would be the actual results, thus indicating that there is no anxiety phenotype. Regardless, the second day of OFA, when a phenotype is shown, also coincides with day 8 post inoculation where the mice are clearly sick, as indicated by the sharp drop in daily food intake that occurs in this model system (see Figure 4F). The measurements used (i.e. Time spent in center and percent time in center) are both measurements that do not take into account the level of activity in these animals and do not normalize for the total level of activity. If these animals are simply sick and do not move much from the sides, it has little to do with anxiety versus sickness behavior which is clearly observed the next day with the great reduction in overall exploratory behavior. Other measurements should be reported to assess anxiety that normalize for activity level and test, such as elevated plus maze, that are most specifically geared for anxiety should be conducted if possible.

3) The authors show interesting data indicating that there is recruitment of cells, including neutrophils, to the brain in tumor bearing animals; however, very little is done in regards to assessing inflammation. They have referenced previous work from their lab (i.e. Braun et al., 2011) to indicate that CNS inflammation can be associated with alterations in muscle proteolysis and demonstrated in that work the impact of *IL-1β* driven neuroinflammation on this process. Assessment of pro-inflammatory cytokines should be reported in this current model system in order to determine if there is actual inflammation that accompanies infiltration.

4) Examination of the impact of oATP administration to block recruitment suggested that neutrophils were the only population that were significantly reduced with oATP treatment. However, although both CD45^high^ myeloid cells and Ly6G^high^ monocytes appeared to be reduced without reaching significance during oATP treatment relative to aCSF tumor controls, the aCSF tumor control animals themselves did not have a significant increase in these populations relative to sham oATP controls. This suggests that there should have been a longer period of time between the cannula placement and tumor inoculation to allow for any inflammation induced by that procedure to resolve. Additionally, it appears that there is really one outlier in both cases that is driving the lack of significance. This is important to clarify because just as CCR2KO significantly lowers the numbers of Ly6C^high^ monocytes as well as neutrophils, the oATP appears to have a similar effect.

5) In order to determine that neutrophils, in general, are impacting the anorexia and muscle loss, a specific treatment targeting neutrophils (e.g. 1A8 antibody mediated depletion), should be conducted.

6) Authors need demonstrate the absence of metastasis in the brains as an event leading to the recruitment of neutrophils.

7) Authors need to compare the immune composition of the brain +/- meninges to clearly confirm their parenchymal location.

8) CCR2 is not the most common pathway known to promote recruitment of neutrophils. What happens when CXCR2 get blocked? The data leading to the selection of CCR2 should be presented.

9) The microglia activation data are not supporting the statement. Authors should present high magnification pictures with quantification of cell body and Sholl analysis to claim that microglia are or are not activated.

[Editors’ note: further revisions were suggested prior to acceptance, as described below.]

Thank you for submitting your article "A distinct neutrophil population invades the central nervous system during pancreatic cancer" for consideration by *eLife*. Your article has been reviewed by three peer reviewers, one of whom is a member of our Board of Reviewing Editors, and the evaluation has been overseen by Tadatsugu Taniguchi as the Senior Editor. The reviewers have opted to remain anonymous.

The reviewers have discussed the reviews with one another and the Reviewing Editor has drafted this decision to help you prepare a revised submission.

Summary:

Dr. Daniel Marks and colleagues reported that the accumulation of circulating myeloid cells in the brain is associated with cachexia symptoms in malignancies outside the central nervous system (CNS), as has been demonstrated in a syngeneic, immunocompetent, mouse model of pancreatic ductal adenocarcinoma (PDAC). They discovered that CD45+CD11b+Ly6G+ neutrophils constitute a major proportion of brain-infiltrating myeloid cells that increases during PDAC progression. Consistently, MPO (a typical component of azurophilic granules in neutrophils and lysosomes in monocytes)-expressing CD45+ globoid cells were found primarily localized in the velum interpositum (VI) layer of meninges surrounding the hippocampus, where Iba1+CD206+ meningeal macrophages are the dominant cellular source of the chemokine CCL2. Pharmacologic blockade of CCR2, but not CXCR2, or genetic deletion of Ccr2 in mouse (not specifically in neutrophils) significantly reduced brain-infiltrating neutrophils, as well as attenuated anorexia and decreased muscle mass loss during PDAC. In addition, intracerebroventricular administration of a P2RX7 antagonist oxidized ATP phenocopied CCR2 blockade in this model, without affecting systemic inflammation or tumor progression. This work suggests a causative link between the accumulation of myeloid cell subpopulations in the brain parenchyma and cancer-associated cachexia.

Essential revisions:

1) An important missing gap is to prove whether neutrophils in the VI, among multiple brain-infiltrating myeloid cell subsets, are the major inducers of cancer-associated anorexia. Further experiments are needed to consolidate whether neutrophils are the only cell involved. For example, antibody-mediated neutrophil depletion (e.g., with Ly6G mAb (clone 1A8), rather than Gr1 mAb) can be a feasible solution. It remains possible that the accumulation of CCR2-expressing monocytes in the CNS, preceding or accompanying the accumulation of neutrophils, also plays a role in the occurrence of cachexia.

2) The authors claim that CCL2 production by meningeal macrophages mediates neutrophil recruitment to the VI layer of meninges, which depends on CCR2, rather than CXCR2 signaling. To consolidate this conclusion, the PDAC model should be tested in genetically modified mice harboring neutrophil-specific deletion CCR2 or CXCR2 (e.g. by the Cre-lox strategy) respectively. As the upregulation of CXCR2 ligands is much greater than CCL2 (Figure 1), and pharmacologic inhibitors may be insufficient to block CXCR2 signaling, CXCR2 deficient mice should be tested in this setting. Although these suggestions can indeed help to nail down the conclusion, it is an overwhelming task to complete all these animal experiments. Therefore, the authors are encouraged to test the essential and doable ones, rephrase their interpretations of the data, as well as improve related discussions.

3) Some interesting observations can be emphasized and discussed. How does PDCA progression initiates CCL2 production by Iba1+CD206+ meningeal macrophages brain-resident macrophages? What are the possible factors (either derived from tumor cells or tumor microenvironment) that specifically activate this cell population? Could brain-infiltrating immune cell populations participate in regulating CCL2 and CXCL2 production in the brain (via positive or negative feedback loops)? Why brain-infiltrating neutrophils show distinct transcriptional features, compared with neutrophils in other compartments.

4) Neutrophil accumulation in the brain in PDCA bearing mice can be blocked with P2RX7 antagonist oATP or CCR2 inhibitor RS504393. Can oATP treatment interfere with CCL2 production by brain resident macrophages, or the CXCL2 level in different brain regions?

5) It is very difficult for readers to identify some labels in the figures. And some of the immunofluorescent images are vague. The quality of the figures must be largely improved.

---

## [Author Response]

[Editors’ note: the authors resubmitted a revised version of the paper for consideration. What follows is the authors’ response to the first round of review.]

1) In order to provide convincing evidence that neutrophils are significantly increased in the CNS, additional flow cytometric data will need to be conducted. As it stands, the gating for CD45highCD11b+ cells may still include microglia, which may also non-specifically bind to antibodies and thus look "positive" for markers (e.g. Ly6G+). Although it is appreciated that the authors attempt to use a bone marrow chimera using chemotherapy based myeloablation (i.e. treosulfan/Ovastat) to circumvent blood brain barrier (BBB) disruptions, the gates in which the GFP+ cells fall in Figure 1—figure supplement 2B could still encompass microglia. Since this is a pivotal point for the paper, it would be crucial to re-conduct some of these experiments utilizing a marker which will eliminate microglia (e.g. Tmem119) to rule out this caveat. This impacts the entire paper since many of the results are based on how they gated for neutrophils in general, identify CCR2+ cells as including a subset of neutrophils in their flow cytometric studies, and sorting for RNAseq is theoretically also utilizing a similar gate.

We performed an additional flow cytometry experiment using the macrophage marker CX3CR1 to identify brain macrophages. Nearly all brain macrophages, including microglia, express CX3CR1, but neutrophils do not. This new supplementary figure is now Figure 2—figure supplement 2. As you can see, 95% of the population we labeled as “microglia” expressed CX3CR1, whereas only 6% of Ly6G+ neutrophils expressed this marker. This confirms the identity of these cells as neutrophils and confirms that they are not microglia. Therefore, all of the neutrophil gates throughout the paper (including the RNASeq) are correct and do not include microglia.

2) Although the authors identify a behavioral phenotype using open field assay (OFA), a two-day test for OFA is highly atypical. There is no training day for OFA, and the results found on day 1 would be the actual results, thus indicating that there is no anxiety phenotype. Regardless, the second day of OFA, when a phenotype is shown, also coincides with day 8 post inoculation where the mice are clearly sick, as indicated by the sharp drop in daily food intake that occurs in this model system (see Figure 4F). The measurements used (i.e. Time spent in center and percent time in center) are both measurements that do not take into account the level of activity in these animals and do not normalize for the total level of activity. If these animals are simply sick and do not move much from the sides, it has little to do with anxiety versus sickness behavior which is clearly observed the next day with the great reduction in overall exploratory behavior. Other measurements should be reported to assess anxiety that normalize for activity level and test, such as elevated plus maze, that are most specifically geared for anxiety should be conducted if possible.

We decided to remove the behavioral analysis data (OFA and Novel Object Test) since they do not contribute significantly to the findings of this manuscript and are challenging to interpret in the context of the decreased locomotor activity associated with cachexia.

3) The authors show interesting data indicating that there is recruitment of cells, including neutrophils, to the brain in tumor bearing animals; however, very little is done in regards to assessing inflammation. They have referenced previous work from their lab (i.e. Braun et al., 2011) to indicate that CNS inflammation can be associated with alterations in muscle proteolysis and demonstrated in that work the impact of IL-1β driven neuroinflammation on this process. Assessment of pro-inflammatory cytokines should be reported in this current model system in order to determine if there is actual inflammation that accompanies infiltration.

We now include an extensive analysis of pro-inflammatory cytokine and chemokine transcripts from the area postrema, hypothalamus, and hippocampus (see new Figure 1). While there is upregulation *Il-1β*, as well as *Ptgs2* (coding for prostaglandin synthase D2), the inflammatory response in the brain during PDAC is dominated by chemokines, suggesting that the cellular infiltration we describe throughout the manuscript is an important aspect of the overall inflammatory response.

4) Examination of the impact of oATP administration to block recruitment suggested that neutrophils were the only population that were significantly reduced with oATP treatment. However, although both CD45high myeloid cells and Ly6Ghigh monocytes appeared to be reduced without reaching significance during oATP treatment relative to aCSF tumor controls, the aCSF tumor control animals themselves did not have a significant increase in these populations relative to sham oATP controls. This suggests that there should have been a longer period of time between the cannula placement and tumor inoculation to allow for any inflammation induced by that procedure to resolve. Additionally, it appears that there is really one outlier in both cases that is driving the lack of significance. This is important to clarify because just as CCR2KO significantly lowers the numbers of Ly6Chigh monocytes as well as neutrophils, the oATP appears to have a similar effect.

We added the following sentence to the Discussion: “It is worth noting that, unlike our previous experiments in non-cannulated mice, we did not observe a significant increase in brain-infiltrating Ly6C^high^ monocytes in aCSF-treated tumor mice compared to oATP-treated sham mice. However, this comparison was nearly significant (p=0.06), and the lack of significance was likely due to variation imposed by the cannulation procedure.”

5) In order to determine that neutrophils, in general, are impacting the anorexia and muscle loss, a specific treatment targeting neutrophils (e.g. 1A8 antibody mediated depletion), should be conducted.

We made numerous attempts to deplete neutrophils using several different techniques. The major issue is that there is massive neutrophil mobilization from the bone marrow during PDAC and the half-life of neutrophils is extremely short (often less than 24 hrs in the circulation), so chronic depletion in this setting is simply not possible. As you can see in Author response image 1, when we administered Ly6G 1A8 antibody daily at a very high concentration (500 ug IP daily), the antibody binds to the Ly6G antigen, as evidenced by the lack of immunoreactivity in subsequent Ly6GPE/ Cy7 labeling. However, there was a suspicious population present that was Ly6C^mid^Ly6G^low^. When we analyzed that population with additional neutrophil markers (Dec205+CD115-) (see Napier et al., 2015). We discovered that the population was in fact neutrophils, and this population was not decreased compared to the isotype control treated animals. We believe that the shift in this population is secondary to surface binding of the detection antibody.

**Author response image 1. sa2fig1:** Neutrophils are not depleted by Ly6G antibody administration. Representative flow cytometry analysis of isolated circulating leukocytes from PDAC-bearing mice, 10 d.p.i. Mice received daily IP injections of 500 ug Ly6G antibody (1A8, Biolegend) starting 2 d.p.i. or isotype control. (**A**) Gating on live, CD45+CD11b+ myeloid cells. Note Ly6G^low^Ly6C^mid^ population in “Tumor Antibody” plot. (**B**) CD115 and Dec205 expression on same CD45+CD11b+ myeloid cells as in A. Population drawn = neutrophils.

We decided not to include these data in the manuscript because they do not contribute to the main conclusions of the study. We also attempted diphtheria toxin-mediated depletion using MRP8^DTR^ mice (by crossing MRP8^Cre^ mice with DTR^fl/fl^ mice, MRP8 is highly specifically expressed in neutrophils). Unfortunately, the dose of DT necessary to maintain chronic depletion was highly toxic to mice, either by the DT itself, or the effects chronic neutrophil depletion (i.e., release of cytotoxic neutrophil granules, etc.).

6) Authors need demonstrate the absence of metastasis in the brains as an event leading to the recruitment of neutrophils.

We performed microscopic analysis of more than a thousand brain sections in this animal model over the last two years. We have never observed even microscopic evidence of metastases in these animals.

7) Authors need to compare the immune composition of the brain +/- meninges to clearly confirm their parenchymal location.

One of the main findings of this manuscript is that neutrophils accumulated in a *specialized layer of meninges* (pia only) located between the hippocampus and thalamus. This layer of meninges cannot be dissected. Furthermore, as demonstrated in Figure 3—figure supplement 2B, the majority of immune cells that infiltrate into the parenchyma are phagocytosed by microglia. Therefore, running flow cytometry on the brain with or without the outer meninges would not significantly contribute to the findings of this manuscript, as it would still include those that are located in VI, which is a substantial percentage of brain-infiltrating immune cells.

8) CCR2 is not the most common pathway known to promote recruitment of neutrophils. What happens when CXCR2 get blocked? The data leading to the selection of CCR2 should be presented.

See new Figure 1 – *Ccl2* is highly upregulated in the hippocampus and is trending toward significance in the hypothalamus (p=0.06) during PDAC. Both *Cxcl1* and *Cxcl2* are upregulated in the hypothalamus and hippocampus. Furthermore, CCL2 is important for monocyte chemotaxis (and important for neutrophil chemotaxis in *certain contexts* – see Souto et al., 2009), while CXCL1 and CXCL2 are usually considered the primary chemokines for neutrophil chemotaxis. Since the brain-infiltrating immune cells we observed in our mouse model of PDAC consisted of neutrophils and monocytes, we decided to target CCR2 and CXCR2 (which is the receptor for CXCL1 and CXCL2). We added a new figure (Figure 4) that describes a series of experiments in which CCR2 and CXCR2 inhibitors are administered to PDAC-bearing mice. Mice that received the CCR2 inhibitor experienced decreased anorexia compared to vehicle-treated mice, whereas mice that received the CXCR2 inhibitor did not. Moreover, we demonstrated a decrease in VI-infiltrating immune cells in mice that received the CCR2 inhibitor, whereas there was no significant change in VI-infiltrating immune cells in mice that received the CXCR2 inhibitor. These data also validate the use of CCR2 knockout mice for more extensive analysis.

9) The microglia activation data are not supporting the statement. Authors should present high magnification pictures with quantification of cell body and Sholl analysis to claim that microglia are or are not activated.

We do not understand the reviewer comment “The microglia activation data are not supporting the statement”, since we performed microglia morphology analysis and observed that there was no microglia activation in the hippocampus in tumor-bearing animals (see Figure 7—figure supplement 2). This is in agreement with our statement “We did not observe any differences in microglia size, Iba-1 staining area, and Iba-1 intensity per cell when comparing aCSF-or oATP-treated tumor animals to oATP-treated sham animals or each other (Figure 7—figure supplement 2).” These results show that microglia activation state in the hippocampus is not affected by the presence of pancreatic tumor or oATP administration to the brain. While we understand and appreciate that there are more advanced means to analyze morphology, the topic of the current manuscript is brain-infiltrating myeloid cells, not resident myeloid cells. However, when we conducted qRT-PCR of hippocampal tissue and demonstrated that there was no difference in expression of transcripts associated with microglia activation, including *Tnf*, *Cd68*, *Tmem119*, *P2y12*, and *P2x7r*, when comparing oATP-treated to aCSF-treated tumor animals. These data are now part of Figure 7—figure supplement 2.

[Editors’ note: what follows is the authors’ response to the second round of review.]

Essential revisions:1) An important missing gap is to prove whether neutrophils in the VI, among multiple brain-infiltrating myeloid cell subsets, are the major inducers of cancer-associated anorexia. Further experiments are needed to consolidate whether neutrophils are the only cell involved. For example, antibody-mediated neutrophil depletion (eg, with Ly6G mAb (clone 1A8), rather than Gr1 mAb) can be a feasible solution. It remains possible that the accumulation of CCR2-expressing monocytes in the CNS, preceding or accompanying the accumulation of neutrophils, also plays a role in the occurrence of cachexia.

We made numerous attempts to deplete neutrophils using the Ly6G clone 1A8 mAb. The major issues are that there is massive neutrophil mobilization from the bone marrow during PDAC, the half-life of neutrophils is extremely short (often less than 24 hrs in the circulation), and PDAC inhibits macrophage phagocytic capabilities (important for Ly6G antibody-mediated neutrophil depletion), so chronic depletion in this setting is simply not possible. We included these data as a new supplemental figure (Figure 3—figure supplement 3). As you can see, when Ly6G 1A8 antibody was administered daily at a very high concentration (500 µg IP daily), the antibody binds to the Ly6G antigen, as evidenced by the lack of immunoreactivity in subsequent Ly6GPE/Cy7 labeling. However, there was a suspicious Ly6C^mid^Ly6G^low^ population present in animals that received the 1A8 antibody. We believe that the shift in this population is secondary to surface binding of the detection antibody. When we analyzed that population with additional neutrophil markers (Dec205+CD115-, see Napier et al., 2015^1^). We discovered that the population was almost exclusively neutrophils. This population was not decreased compared to the isotype control treated animals.

To comment on this issue, we added the following to the Discussion: “We were unable to deplete neutrophils, despite daily IP injections of high dose anti-Ly6G antibody. […] Moreover, previous studies demonstrated that PDAC reprograms macrophages, effectively impairing their phagocytic capabilities^5,6^, Since macrophage phagocytosis was shown to be required for Ly6G antibody-mediated neutrophil depletion^7^, this may explain why the Ly6G antibody bound to neutrophils, but they were not depleted.”

We were in the process of conducting additional experiments to more thoroughly address this comment, but these were stopped as a result of the COVID outbreak. Since we were unable to definitively identify neutrophils as they key cell type involved in cachexia, we modified our conclusions. We renamed the manuscript “Circulating myeloid cells invade the central nervous system to mediate cachexia during pancreatic cancer.” We also rephrased the Discussion to read as “Third, since we were not able to selectively deplete neutrophils, we cannot definitely conclude that these cells are the sole cellular drivers of cachexia during PDAC. […] However, there were far fewer Ly6C^high^ monocytes (≈2,000) in the brain than neutrophils (≈9,000) during PDAC, and these cells only constituted about 15% of brain CD45^high^ myeloid cells (vs. approximately 50% for neutrophils).”

2) The authors claim that CCL2 production by meningeal macrophages mediates neutrophil recruitment to the VI layer of meninges, which depends on CCR2, rather than CXCR2 signaling. To consolidate this conclusion, the PDAC model should be tested in genetically modified mice harboring neutrophil-specific deletion CCR2 or CXCR2 (e.g. by the Cre-lox strategy) respectively. As the upregulation of CXCR2 ligands is much greater than CCL2 (Figure 1), and pharmacologic inhibitors may be insufficient to block CXCR2 signaling, CXCR2 deficient mice should be tested in this setting. Although these suggestions can indeed help to nail down the conclusion, it is an overwhelming task to complete all these animal experiments. Therefore, the authors are encouraged to test the essential and doable ones, rephrase their interpretations of the data, as well as improve related discussions.

*Ccr2* floxed animals are not commercially available, nor were we able to find any publications describing these animals. *Cxcr2* floxed animals are commercially available, but take 10-14 weeks for cryorecovery, meaning that even in ideal circumstances it will take at least one year for breeding (with MRP8^Cre^ mice, which is the only neutrophil specific Cre line available) and generating the appropriate experimental animals. Moreover, CXCR2 is almost exclusively expressed on neutrophils, making it unnecessary to selectively delete the associated gene from neutrophils. *Cxcr2* knockout animals are commercially available, but only as heterozygotes, which would have required at least 4 months to generate experimental animals prior to the COVID-19 outbreak. We ordered these animals and were planning to initiate breeding, but since all breeding and animal orders have been halted, we will not be able to have experimental animals available for at least eight months.

However, the CXCR2 blockade with the dosing regimen we used was previously validated^8^. Moreover, in response to the comment “As the upregulation of CXCR2 ligands is much greater than CCL2 (Figure 1)”, we respectfully disagree. As Figure 1 shows, in the hypothalamus *Ccl2* is upregulated approximately two-fold, *Cxcl1* is upregulated approximately five-fold, and *Cxcl2* is upregulated approximately 15-fold. However, as Figure 3 shows, neutrophils *donot* infiltrate the hypothalamus. More importantly, in the dissected hippocampal blocks, which included the VI, where neutrophils *do* infiltrate, *Ccl2* is upregulated 17-fold, whereas *Cxcl1* is upregulated 6-fold and *Cxcl2* is upregulated 12-fold.

Nevertheless, we appreciate the reviewer’s concerns and rephrased our interpretations of the data and improved related discussions. We changed the associated section in the Results to read “These data demonstrate that pharmacologic inhibition of CCR2 is important for cachexia and immune cell recruitment to the brain during PDAC.” In addition, we added the following to the Discussion: “Also, it is worth noting, that our pharmacologic inhibition of CXCR2 may have been incomplete. Therefore, it remains to be determined whether CXCR2 ligands are important for recruitment of cachexia-inducing neutrophils to the CNS.”

3) Some interesting observations can be emphasized and discussed. How does PDCA progression initiates CCL2 production by Iba1+CD206+ meningeal macrophages brain-resident macrophages? What are the possible factors (either derived from tumor cells or tumor microenvironment) that specifically activate this cell population? Could brain-infiltrating immune cell populations participate in regulating CCL2 and CXCL2 production in the brain (via positive or negative feedback loops)? Why brain-infiltrating neutrophils show distinct transcriptional features, compared with neutrophils in other compartments.

We conducted an experiment demonstrating that tumor-conditioned media induces chemokine induction in microglia in vitro. This experiment involved isolating mixed glia from one to three-day old mouse pups, then after 14-16 days we removed microglia through shaking and treated them with KPC-conditioned media. 16 hours after treatment, we extracted RNA and performed qRT-PCR tumor-bearing mice (per Figure 1). We included these data as a new main figure (now Figure 6). As this figure shows, *Ccl2,Cxcl1*, and *Cxcl2* were all upregulated in microglia by tumor-conditioned media, but *Cxcl9* and *Cxcl10* were not. Of these transcripts, *Ccl2* was by far the most highly upregulated (136-fold vs. 15-fold for *Cxcl1* and 46-fold for *Cxcl2*) This recapitulates our in vivo data. We also included a positive control experiment, demonstrating nonselective robust upregulation of all chemokines analyzed in response to LPS treatment (Figure 6C). These findings demonstrate that tumor-derived factors cause brain macrophages to express chemokines, specifically CCL2, which in turn recruits neutrophils and other myeloid cells to the CNS. We understand this does not perfectly model meningeal macrophages; these cells cannot be isolated at sufficient numbers to perform a similar experiment. Furthermore, it was previously demonstrated that microglia likely become meningeal macrophage-like when they are in culture (higher expression of CD44, CD68)^9^.

In regards to the comment “Could brain-infiltrating immune cell populations participate in regulating CCL2 and CXCL2 production in the brain (via positive or negative feedback loops)”, these findings indicate it is unlikely that brain-infiltrating immune cell populations participate in regulating CCL2 and CXCL2 production in the brain (via positive or negative feedback loops).

In regards to the comment “Why brain-infiltrating neutrophils show distinct transcriptional features, compared with neutrophils in other compartments”, we believe this is due to the unique microenvironment of the CNS compared to other organs. It was previously demonstrated that the CNS induces a more “inflammatory” transcriptional profile in infiltrating myeloid cells compared to those that infiltrate other organs^10^. Therefore, we added the following to the Discussion: “We observed that the transcriptional profile of brain-infiltrating neutrophils was distinct from those in the circulation, liver, and tumor. […] Reasons for this are not entirely clear but may be due to relative lack of regulatory T-cells, the blood brain barrier preventing access to soluble anti-inflammatory factors, and presence of immunogenic substances more abundant in the brain, such as myelin-associated lipids.”

4) Neutrophil accumulation in the brain in PDCA bearing mice can be blocked with P2RX7 antagonist oATP or CCR2 inhibitor RS504393. Can oATP treatment interfere with CCL2 production by brain resident macrophages, or the CXCL2 level in different brain regions?

We performed an additional qRT-PCR experiment investigating *Ccl2*, *Cxcl1*, and *Cxl2* transcript expression in the hypothalamus and hippocampus in oATP-treated tumor-bearing animals compared to aCSF-treated tumor-bearing animals at 10 d.p.i. There was no difference in expression of these transcripts with oATP treatment. However, this was from whole tissue, so brain macrophages were not isolated. PCR may not be sensitive enough to pick up these differences. So as to not unduly burden the reader, we chose not to include these data in the manuscript. Prior to the COVID-19 outbreak, we were planning on conducting an in vitro experiment to address this comment but were unable to perform it before labs were ordered shut down.

5) It is very difficult for readers to identify some labels in the figures. And some of the immunofluorescent images are vague. The quality of the figures must be largely improved.

The pdf file with images embedded was highly compressed to reach the appropriate size for upload. The embedded images therefore have significantly compromised quality and are placed within the manuscript to make references to figures easier. We encourage the reviewers to refer to the separate uploaded pdf figure files, which are significantly increased quality/resolution. In addition, if the manuscript is accepted, we will upload high-resolution pdfs, which will further improve image quality. However, we made significant changes to the figures, which improved their quality substantially. Labels are much easier to identify, and the immunofluorescent images are less vague (especially Figures 3, 4, 5, 7, and 8).

References

1. Napier RJ, Norris BA, Swimm A, et al. Low Doses of Imatinib Induce Myelopoiesis and Enhance Host Anti-microbial Immunity. PLOS Pathogens 2015; 11(3): e1004770.

2. Daley JM, Thomay AA, Connolly MD, Reichner JS, Albina JE. Use of Ly6G-specific monoclonal antibody to deplete neutrophils in mice. Journal of leukocyte biology 2008; 83(1): 64-70.

3. Reber LL, Gillis CM, Starkl P, et al. Neutrophil myeloperoxidase diminishes the toxic effects and mortality induced by lipopolysaccharide. Journal of Experimental Medicine 2017; 214(5): 1249-58.

4. Pollenus E, Malengier-Devlies B, Vandermosten L, et al. Limitations of neutrophil depletion by anti-Ly6G antibodies in two heterogenic immunological models. Immunology Letters 2019; 212: 30-6.

5. Liu M, O’Connor RS, Trefely S, Graham K, Snyder NW, Beatty GL. Metabolic rewiring of macrophages by CpG potentiates clearance of cancer cells and overcomes tumor-expressed CD47−mediated ‘don’t-eat-me’ signal. Nature Immunology 2019; 20(3): 265-75.

6. Mantovani A, Sica A. Macrophages, innate immunity and cancer: balance, tolerance, and diversity. Current opinion in immunology 2010; 22(2): 231-7.

7. Bruhn KW, Dekitani K, Nielsen TB, Pantapalangkoor P, Spellberg B. Ly6G-mediated depletion of neutrophils is dependent on macrophages. Results Immunol 2015; 6: 5-7.

8. Nywening TM, Belt BA, Cullinan DR, et al. Targeting both tumour-associated CXCR2^+^ neutrophils and CCR2^+^ macrophages disrupts myeloid recruitment and improves chemotherapeutic responses in pancreatic ductal adenocarcinoma. Gut 2018; 67(6): 1112.

9. Bohlen CJ, Bennett FC, Tucker AF, Collins HY, Mulinyawe SB, Barres BA. Diverse Requirements for Microglial Survival, Specification, and Function Revealed by Defined-Medium Cultures. Neuron 2017; 94(4): 759-73.e8.

10. Spath et al., 2017 S, Komuczki J, Hermann M, et al. Dysregulation of the Cytokine GM-CSF Induces Spontaneous Phagocyte Invasion and Immunopathology in the Central Nervous System. Immunity 2017; 46(2): 245-60.